


# A WRF-Chem study on the variability of $CO_2$, $CH_4$ and CO concentrations at Xianghe, China supported by ground-based observations and TROPOMI

Sieglinde Callewaert[1], Minqiang Zhou[1,2], Bavo Langerock[1], Pucai Wang[2], Ting Wang[2],
Emmanuel Mahieu[3], and Martine De Mazière[1]

[1]Royal Belgian Institute for Spacy Aeronomy (BIRA-IASB), Brussels, Belgium
[2]CNRC & LAGEO, Institute of Atmospheric Physics, Chinese Academy of Sciences, Beijing, China
[3]UR SPHERES, Department of Astrophysics, Geophysics and Oceanography, University of Liège, Liège, Belgium
**Correspondence:** Sieglinde Callewaert (sieglinde.callewaert@aeronomie.be)

**Abstract.** The temporal variability of both surface concentrations and column abundances of $CO_2$, $CH_4$ and CO at the Xianghe site in China are analyzed with the Weather Research and Forecast model coupled with Chemistry (WRF-Chem). Simulations of these in situ (PICARRO) and remote sensing (TCCON-affiliated) measurements are produced by the model's passive tracer option, called WRF-GHG, from September 2018 until September 2019. Our analysis found a good model performance with

correlation coefficients between observations and simulations up to 0.85 for $CO_2$ and 0.69 for CO. Key source sectors for every gas are revealed by tracking the anthropogenic fluxes in separate tracer fields. While there are slight variations in the relative impacts of these source sectors between surface and column observations, owing to differences in the sensitivity footprint of each observation type, the primary sectors influencing the various species are evident. For $CO_2$ the industry, energy and biosphere sectors are found to be the primary contributors to the total simulated concentration, whereas $CH_4$ concentrations are

predominantly attributed to the energy, agriculture and residential & waste sectors. For CO, industry is the largest contributing sector at Xianghe, followed by residential and transportation sources. Differences among the various observation types were particularly visible in the contributions of the biosphere to $CO_2$ and the energy sector to $CH_4$, as their largest sources are located further away from Xianghe. Further, the influence of meteorological factors on the variability observed in the different time series was analyzed. We found that southwest winds typically bring polluted air masses from the North China Plain to the

site, while northern winds are associated with cleaner conditions. Variability in surface measurements is primarily driven by the daily cycle of accumulation and atmospheric mixing linked with the planetary boundary layer height. Furthermore, the study demonstrates the ability to detect strong regional sources at Xianghe depending on wind direction. To address inconsistencies between the simulations and observations of $CH_4$, we looked at TROPOspheric Monitoring Instrument (TROPOMI) satellite observations. We found that the model underestimation of $CH_4$ in summer and overestimation in winter may result from a

combination of a similar bias in the lateral boundary conditions and an incorrect monthly variation of the $CH_4$ emissions in the agriculture and/or waste sectors of the CAMS-GLOB-ANT inventory over north China. Additionally, WRF-GHG simulations indicated a possible overestimation of coal mine emissions nearby Tangshan, which could not be confirmed nor contradicted by the TROPOMI observations. In summary, our findings highlight the value of WRF-GHG to interpret both surface and column





observations at Xianghe, offering source sector attribution and insights in the link with local and large-scale winds based on the simultaneously computed meteorological fields. However, given the long lifetime of the considered species and the fact that WRF-GHG is a regional model, accurate initial and lateral boundary conditions remain crucial. The dependence on precise input emission data on the other hand, can be used to evaluate the existing bottom-up inventories.

## 1    Introduction

Carbon dioxide ($CO_2$) is the most important anthropogenic greenhouse gas (GHG) and is therefore a key player in climate change. Driven by human activities, atmospheric $CO_2$ has been increasing since the pre-industrial era to a level that is higher than ever (Masson-Delmotte et al., 2021). Methane ($CH_4$) is the second largest anthropogenic contributor to global warming and overall its global concentration has also been rising the last 200 years to levels that are above the natural changes of the last millennia (Masson-Delmotte et al., 2021). Moreover $CH_4$ has a 28 times larger global warming potential over a period of 100 year and a 10 times shorter atmospheric lifetime, compared to $CO_2$. Controlling $CH_4$ emissions is therefore a priority to mitigate climate change in the near future (Saunois et al., 2020).

Because of rapid industrialization in the past decades and its heavy dependence on coal, China is the world's largest emitter of $CO_2$ and $CH_4$ (Friedlingstein et al., 2022; Worden et al., 2022). The main anthropogenic $CO_2$ sources in China are industry, power generation, residential and commercial activities and transportation (Zhao et al., 2012), while sectors such as coal mining, livestock, rice paddies, landfills and wastewater management are the largest contributors to the $CH_4$ emissions in China (Chen et al., 2022). China has pledged to reach its carbon peak by 2030 and neutrality by 2060. To help battle climate change and reach these goals, it is essential to have accurate observations of the GHG concentrations. Not only does atmospheric monitoring aid in revealing sources and sinks and controlling the impact of mitigation measures, but by studying temporal variations a better understanding of the carbon cycle and its interactions with the atmosphere can be achieved.

Since 2018, both ground-based in situ and remote sensing observations of GHGs have been deployed at the Xianghe observatory, which is located about 50 km southwest of Beijing. Its location in the center of the Beijing-Tianjin-Hebei (BTH) megalopolis makes it an interesting site to study the properties and variability of GHGs in a polluted area. The remote sensing observations are made by a Fourier Transform Infrared (FTIR) spectrometer and are part of the international Total Column Carbon Observing Network (TCCON), while the in situ concentrations are measured by a PICARRO cavity ring-down spectroscopy (CRDS) analyzer that is installed on a tower at an altitude of 60 m above the ground. Some first insights in the observed time series were made by Yang et al. (2020, 2021) and Ji et al. (2020). The seasonal cycle of $CO_2$ at Xianghe is consistent with other sites, with larger values in winter and lower in summer, driven by an increase of fossil fuel from traffic and heating systems in winter and an uptake by the biosphere in summer due to photosynthesis (Yang et al., 2020, 2021). The $CH_4$ seasonal cycle however is different from elsewhere, with larger concentrations in summer and autumn and lower values in spring (Yang et al., 2020; Ji et al., 2020). Furthermore, the column observations of $CO_2$, $CH_4$ and CO show a large day-to-day variability and are correlated with each other. Yang et al. (2020) showed that the high values are related to both local pollution and pollution originating from the south, while low concentrations are corresponding with clean airmasses from more remote





regions in the north.

This work aims to perform a comprehensive analysis of both in situ and column observations of $CO_2$, $CH_4$ and additionally CO at Xianghe to complement previous studies and gain a better understanding of the causes of the observed temporal variabilities. To achieve this goal, we will simulate the time series at a high spatial resolution with the WRF-Chem model for greenhouse gases (WRF-GHG). This widely used regional atmospheric transport model simulates the 3-D concentrations together with meteorological fields without chemical interactions, which is a valid assumption regarding the regional domain and the relatively long atmospheric lifetime of the target species ($\sim 100$ yrs for $CO_2$, $\sim 10$ yrs for $CH_4$ and several weeks for CO)(Dekker et al., 2017). WRF-GHG has already shown to be a useful tool to study $CO_2$ fluxes and variability in China (Dayalu et al., 2018; Liu et al., 2018; Li et al., 2020; Dong et al., 2021). However, and to our best knowledge, applications to $CH_4$ or CO observations in China have not been reported yet. Elsewhere, this model was successfully used to analyze comparable observations (Zhao et al., 2019; Hu et al., 2020; Park et al., 2020; Callewaert et al., 2022). Therefore, this study will additionally assess the model's capability of simulating these time series in north China and highlight its strengths and weaknesses in this region.

The work is structured as follows: in Sect. 2 the Xianghe site and its observations are described, together with the $XCH_4$ product of TROPOMI (the TROPOspheric Monitoring Instrument onboard Sentinel-5P), which will give additional insight into the results. It is followed in Sect. 3 by an overview of the WRF-GHG model set-up, input data and sensitivity tests. Section 4 describes how the TROPOMI data is compared with the model output. The model performance at Xianghe is discussed in Sect. 5, followed in Sect. 6 by an assessment of the different source sectors that impact the Xianghe observations. An detailed analysis on the found seasonal $CH_4$ bias is given in Sect. 7. Section 8 describes the different factors influencing the observations at Xianghe and finally, the conclusions are drawn in Sect. 9.

## 2 Description of site and observational data sets

### 2.1 Xianghe

The observation site is situated in Xianghe county (39.75° N, 116.96° E; 30 m a.s.l.), a suburban area in the Beijing-Tianjin-Hebei (BTH) region in north China. The center of Xianghe is about 2 km to the east of the site, while the metropolitan cities of Beijing and Tianjin are located about 50 km to the northwest and 70 km to the south-southeast, respectively (see Fig. 1b). Cropland and irrigated cropland are the predominant kind of vegetation in the area. The East Asian Monsoon, which causes hot, humid summers with plenty of precipitation and cold, dry winters, determines the climate.

Since 1974, atmospheric observations are made at the Xianghe observatory by the Institute of Atmospheric Physics (IAP), Chinese Academy of Sciences (CAS). In June 2016 a FTIR spectroscopy instrument (Bruker IFS 125HR) was installed on the roof of the observatory, two years later, a solar tracker was added to the setup and continuous measurements are made from June 2018 onwards. This ground-based remote sensing instrument measures spectra in the infrared and is affiliated with TCCON (Wunch et al., 2011; Zhou et al., 2022), providing total column-averaged dry air mole fractions of $CO_2$, $CH_4$ and CO (the so-called Xgas). The measurement uncertainty is about 0.6 ppm for $XCO_2$, 6 ppb for $XCH_4$ and 2 ppb for XCO. Further details about the instrument and retrieval methodology can be found in Yang et al. (2020).



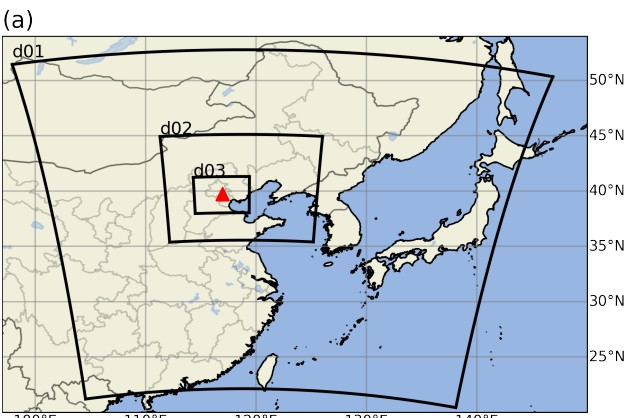
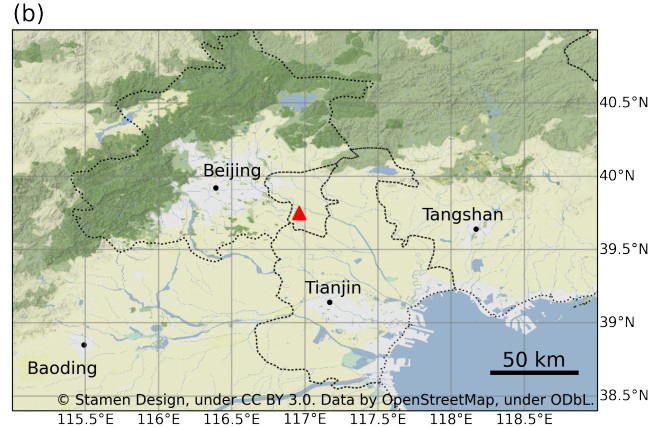

**Figure 1.** (a) Location of the WRF-GHG domains, with horizontal resolutions of 27 km (d01), 9 km (d02) and 3 km (d03). All domains have 60 (hybrid) vertical levels extending from the surface up to 50 hPa. (b) Terrain map including the largest cities in the region of Xianghe, roughly corresponding to d03. The location of the Xianghe site is indicated by the red triangle in both maps.

Additionally, in situ mole fractions of $CO_2$ and $CH_4$ are measured by a PICARRO cavity ring-down spectroscopy G2301 analyzer since June 2018. The instrument is installed on a tower at 60 m above the ground. More detail about the measurement setup is given in Yang et al. (2021). The measurement uncertainty is 0.06 ppm for $CO_2$ and 1 ppb for $CH_4$. There are no in situ observations of CO available at Xianghe.

## 2.2 TROPOMI

The TROPOMI instrument on board the Sentinal-5 Precursor (S5P) satellite is observing the Earth on a polar sun-synchronous orbit. With a daily global coverage, it measures solar backscatter in the near and shortwave infrared absorption bands of which column-average mixing ratios of $CH_4$ can be retrieved. In our study, we use daily and monthly mean L3 values on a 0.1° rectangular lat-lon grid, which were post-processed at BIRA-IASB using the HARP toolset, from the the bias-corrected reprocessed L2 RemoTec-S5P $XCH_4$ product from SRON, where a quality filter of 1.0 was applied.

The L2 product was evaluated at Xianghe by Yang et al. (2020) and Tian et al. (2022): they found a small negative bias of -0.6% and -0.39% with TCCON $XCH_4$, respectively. These values are well within the mission requirements of 1.5 % and demonstrate the great quality of TROPOMI $XCH_4$ in this part of China.

## 3 WRF-GHG modelling system

We use the Weather Research and Forecasting model coupled with Chemistry version 4.1.5 (WRF-Chem, Grell et al. (2005);
Skamarock et al. (2019); Fast et al. (2006)) in its passive tracer option, called WRF-GHG (Beck et al., 2011). WRF-GHG is a Eulerian atmospheric transport model that simulates the 3-D concentration of trace gases at every time step simultaneously with meteorological fields. The model configuration consists of three nested domains with increasing resolution in a Lambert





Conformal projection (see Fig. 1a). The parent domain (d01) has 134 by 130 grid cells of $27\times27$ km$^2$ and covers a large part of China, Mongolia, North and South Korea and Japan. The second domain (d02), which has 133 by 121 grid cells of

$9\times9$ km$^2$, mainly covers north China. Finally, the innermost domain (d03) has a resolution of $3\times3$ km$^2$ over 145 by 124 grid cells and almost completely covers BTH. There are 60 vertical levels between the surface and 50 hPa. The initial set of physical parameterization schemes was taken from Li et al. (2020) and Dong et al. (2021) as they have shown good model performance for simulating $CO_2$ concentrations in China. However, some alternative schemes were tested for the longwave and shortwave radiation, planetary boundary layer (PBL) and surface layer physics. More detail on these sensitivity tests and

the final configuration is given in Sect. 3.2.

## 3.1    Input data

The model was driven by the hourly European Centre for Medium-Range Weather Forecasts (ECMWF) global ERA5 reanalysis data set ($0.25° \times 0.25°$, Hersbach et al. (2023a, b)) for meteorological fields. The concentration fields for $CO_2$ and $CH_4$ are initialized by the 3-hourly Copernicus Atmosphere Monitoring Service (CAMS) global reanalysis for greenhouse gases

(EGG4), while the 6-hourly reactive gases product is used for CO (EAC4, Inness et al. (2019)). These CAMS reanalysis data sets are also used at the model domain boundaries to represent influences coming from outside the parent domain (d01). The evolution of these initial and lateral boundary conditions inside the domain over time is stored in a separate tracer, the so-called background tracer. Similarly, the evolution of concentrations caused by emissions within the boundaries of d01 is saved in different tracers, dependent on their source sector. The sum of all tracers, including the background, gives the total simulated

concentrations which can be compared to the observations.

Because of the large anthropogenic activity in the study region, the choice of anthropogenic flux inventory is likely important for the accuracy of the simulations. Therefore several options were tested for their capability in simulating the different observations at the Xianghe site. An overview of which inventories were considered and what the final choice is, can be found in the next section (Sect. 3.2). Further, biomass burning emissions are coming from the Fire INventory from NCAR (FINN

v2.5, Wiedinmyer and Emmons (2022)) for all species. The observation-based global $pCO_2$ climatology from Landschützer et al. (2017) is used to represent the ocean-atmosphere exchange of $CO_2$, while the $CH_4$ fuxes from wetlands are taken from the WetCHARTS v1.0 climatology (Bloom et al., 2017). Finally, WRF-GHG calculates the biogenic $CO_2$ fluxes online based on the Vegetation Photosynthesis and Respiration Model (VPRM, Mahadevan et al. (2008); Ahmadov et al. (2007)). It uses its own calculated 2 m temperature and downward shortwave radiation together with surface reflectance data from the Moder-

ate Resolution Imaging Spectroradiameter (MODIS) onboard the Aqua and Terra satellites. The extra required parameters for VPRM are taken from Li et al. (2020).

## 3.2    Sensitivity tests

As explained in the previous sections, several physical parameterization options and anthropogenic flux inventories were tested against the observations. Table 1 gives an overview of the five physics combinations that were tested. Note that test E has

exactly the same parameterization set as in the previously mentioned studies (Li et al., 2020; Dong et al., 2021). Further, the





model code was adapted to include different anthropogenic emission inventories in separate tracers. As such, one simulation is sufficient to compare the effect of using different inventories. The anthropogenic flux inventories tested are the following: EDGAR GHG v6.0 (for $CO_2$ and $CH_4$, Ferrario et al. (2021)), EDGAR Air Pollutants v5.0 (for CO, Crippa et al. (2019)), CAMS-GLOB-ANT v5.3 (for $CO_2$, $CH_4$ and CO, Granier et al. (2019); Soulie et al. (2023)), PKU v2 (for $CO_2$ and CO, Wang

et al. (2013); Zhong et al. (2017)), REAS v3.2.1 (for $CO_2$ and CO, Kurokawa and Ohara (2020)), MEICv3.1 (for $CO_2$ and CO, http://www.meicmodel.org/ ), ODIAC2020b (for $CO_2$, Oda and Maksyutov (2011); Oda and Maksyuto (2015); Oda et al. (2018)) and FFDAS v2.2 (for $CO_2$, Asefi-Najafabady et al. (2014)). Monthly fluxes are disaggregated into hourly fluxes using the temporal factors of Crippa et al. (2020), Guevara et al. (2021) and Nassar et al. (2013).

The five simulations, representing different combinations of physical parameterization schemes and anthropogenic fluxes, were

run over three periods of about 2 weeks spread over the year: 1-17 October 2018, 1-17 February 2019 and 10-25 June 2019. The first 48h were regarded as spin-up and are not taken into account in the analysis.

The model cell which covers the location of the instrument is selected to compare with the in situ observations. Because the concentrations are measured at an altitude of 60 m.a.g.l., the WRF-GHG profile is interpolated to this altitude, using the model surface as ground level. Finally, the observations are averaged over a period of 30 minutes around the hourly model output. The

same model cell is used to compare with the column observations. The five TCCON observations that are closest in time with the WRF-GHG output, but deviate no more than 15 minutes, are averaged and used for the comparison. The model profile is extended above 50 hPa with the TCCON a priori profile and then smoothed by using the averaging kernels in order to account for the instrument and retrieval characteristics (Rodgers and Connor, 2003).

For each time series the root mean square error (RMSE), mean bias error (BIAS) and Pearson correlation coefficient (CORR)

were calculated. In order to find the most suitable combination of physical parameterization schemes and anthropogenic emission inventory, a combined skill score (S) was computed as follows, based on Gbode et al. (2019):

$$S = (1 - RMSE_{norm}) + (1 - |BIAS_{norm}|) + CORR_{norm}, \tag{1}$$

where $X_{norm} = \frac{X_i - X_{min}}{X_{max} - X_{min}}$ is the normalized statistical metric. As such, the combination with the highest S will overall have the lowest RMSE, lowest absolute BIAS and highest CORR. Exact values of the statistical metrics and combined skill

scores for every sensitivity test can be found in Appendix A. Unfortunately, the best combination of physical parameterization scheme and anthropogenic flux inventory is different among the five different time series (surface and column $CO_2$ and $CH_4$, and column CO). Therefore, the final combination was determined through a logical process, which is outlined in Appendix A. Whenever necessary, preference was given to the surface data, as it is assumed that the physical schemes will have the highest impact on these simulations. This approach lead to the settings of test B, together with CAMS-GLOB-ANT v5.3 fluxes for

$CO_2$ and $CH_4$ and REAS v3.2.1 (Regional Emission Inventory in Asia) fluxes for CO. Table 2 shows the final set of physical parameterization schemes.

Remark that both chosen anthropogenic inventories additionally provide sector-specific information. We decided to include this information in our simulations by linking different sectors to separate tracers. The 11 sectors from CAMS-GLOB-ANT were aggregated into five broad sectors to not make the model simulations computationally too expensive. A similar aggregation was





| Test | PBL | Surface Layer | Radiation |
|------|-----|---------------|-----------|
| A | YSU scheme (option 1) | Revised MM5 scheme (option 1) | RRTMG (option 4) |
| B | MYJ scheme (option 2) | Eta similarity scheme (option 2) | RRTMG (option 4) |
| C | MYNN3 scheme (option 6) | Eta similarity scheme (option 2) | RRTMG (option 4) |
| D | MYNN3 scheme (option 6) | Revised MM5 scheme (option 1) | RRTMG (option 4) |
| E | YSU scheme (option 1) | Revised MM5 scheme (option 1) | RRTM and Dudhia (option 1) |

**Table 1.** Overview of sensitivity tests on different physical parameterization options. They are a combination of three different PBL schemes: Yonsei University (Hong et al., 2006), Mellor-Yamada-Janjic (Janjić, 1994) and Mellor-Yamada-Nakanishi Niino Level 3 (Nakanishi and Niino, 2006, 2009; Olson et al., 2019); two surface layer schemes: Revised MM5 (Jiménez et al., 2012) and Eta similarity (Janjić, 1994); and two radiation schemes: RRTMG Longwave and Shortwave schemes (Iacono et al., 2008) versus RRTM Longwave and Dudhia Shortwave schemes (Dudhia, 1989; Mlawer et al., 1997).

| Physics | Scheme name | Option |
|---------|-------------|--------|
| Microphysics | Morrison 2-moment | 10 |
| Longwave radiation | RRTMG | 4 |
| Shortwave radiation | RRTMG | 4 |
| Planetary boundary layer | Mellor-Yamada-Janjic | 2 |
| Surface layer | Eta similarity | 2 |
| Cumulus | Grell 3D Ensemble | 5 |
| Land surface | Unified Noah Land Surface Model | 2 |

**Table 2.** Overview of physical parameterization options used for final WRF-GHG simulations.

performed on the REAS sectors. The mapping is given in Table 3. This will allow us to track the respective contributions to the total simulated concentrations of the following sources: energy, industry, transportation, residential & waste and agriculture. More detail about what is included in every sub-sector can be found in the documentation of the respective data set.

## 4  Comparing TROPOMI with WRF-GHG

To compare the spatial XCH$_4$ distribution of TROPOMI with those of WRF-GHG, XCH$_4$ values are calculated from the hourly
3-D model output data as follows:

$$XCH_4 = \frac{\sum_i \nu_i \rho_i^{da} \tau_i}{\sum_i \rho_i^{da} \tau_i}, \text{ with } \rho_i^{da} = \frac{P_i}{RT_i} \frac{1}{1 + 1.6075 q_i}. \tag{2}$$

In the above equation $\nu_i$ is the CH$_4$ volume mixing ratio (ppb), $\rho_i^{da}$ the dry air number density (mol m$^{-3}$) and $\tau_i$ the thickness of layer $i$ (m). The dry air number density $\rho_i^{da}$ is calculated according to the ideal gas law, where $P_i$, $T_i$ and $q_i$ are the air pressure (Pa), temperature (K) and water vapour mixing ratio (kg kg$^{-1}$) in WRF-GHG layer $i$, respectively. Finally, $R$ is the
ideal gas constant 8.3145 J K$^{-1}$ mol$^{-1}$.





| CAMS-GLOB-ANT (for $CO_2$ and $CH_4$) | This study | REAS (for CO) |
|---|---|---|
| Power generation (ene) | | Power plants point |
| Fugitives (fef) | Energy | Power plants non-point |
| Oil refineries and transformation sector (ref) | | |
| Industrial processes (ind) | Industry | Industry |
| Road transportation (tro) | | Road transport |
| Off Road transportation (tnr) | Transport | Other transport |
| Ships (shp) | | |
| Residential, commercial and other combustion (res) | Residential & | Domestic |
| Solid waste and waste water (swd) | Waste | |
| Agriculture soils (ags) | | |
| Agricultural waste burning (awb) | Agriculture | |
| Agriculture livestock (agl) | | |

**Table 3.** Overview of mapping between the emission sectors provided by CAMS-GLOB-ANT v5.3 (first column) and REAS v3.2.1 (third column) and the five broad sectors used in this study (second column).

TROPOMI has a local overpass time of around 13:30, so we compute the simulated daily mean $XCH_4$ by taking the average over 12h-15h LT. To take into account the large day-to-day variability in the spatial coverage of the TROPOMI product due to changing cloud cover ($XCH_4$ is only retrieved on cloud free pixels), we regrid the TROPOMI L3 daily weights (which are a measure of how much information is in each of the L3 grid cells) to the WRF-GHG model resolution and apply this to the

simulated daily means. As such, the model data at locations where there is no observation will not be included on that day. The monthly and seasonal means are then computed as weighted averages of the daily information.

We will compare the TROPOMI data with WRF-GHG $XCH_4$ in domain 2 which has a horizontal resolution of $9\times9$ km$^2$ as this is very close to the $0.1°$ resolution of TROPOMI and allows us to make an evaluation on a larger region than only Beijing and Tianjin. We did not apply a smoothing of the model profiles with the TROPOMI averaging kernels as we only want to make a

qualitative comparison here and focus on the spatial gradients of both products. Remark that in the corresponding figures, we have shifted the colour scale of the WRF-GHG concentrations to be 40 ppb higher than those of TROPOMI to have both maps in comparable colors. The value of 40 ppb should compensate for the model top of WRF-GHG being at 50 hPa, which is lower than TROPOMI.

## 5  Model performance

With the model settings as elaborated in Sect. 3, WRF-GHG was run from 15 August 2018 to 1 September 2019. However, the first two weeks were regarded as a spin-up phase, so the analysis is made on data of one full year: from 1 September 2018 until 1 September 2019. This conservative spin-up period is implemented to ensure thorough mixing of the tracers within the



|  | CO$_2$ (ppm) | | CH$_4$ (ppb) | | CO (ppb) |
| --- | --- | --- | --- | --- | --- |
|  | surface | column | surface | column | column |
| BIAS | 2.52 | -1.43 | 9.43 | -3.03 | 0.42 |
| RMSE | 20.94 | 2.45 | 251.82 | 23.96 | 31.85 |
| CORR | 0.68 | 0.85 | 0.56 | 0.56 | 0.69 |

**Table 4.** Statistics for the different observations at Xianghe site: mean bias error (BIAS), root mean square error (RMSE) and Pearson correlation coefficient (CORR).

domain. The complete data set can be accessed on https://doi.org/10.18758/P34WJEW2 (Callewaert, 2023). The WRF-GHG output was compared with the observations in the same way as for the sensitivity tests (see Sect. 3.2).

An overview of the simulated and observed time series of the GHG concentrations at Xianghe is shown in Fig. 2, while Fig. 3 shows the differences between simulated and observed data. In general we find that WRF-GHG is quite accurate in simulating these measurements: the XCO$_2$ observations are slightly underestimated with a mean bias error of -1.43 ppm (see Table 4), while the surface CO$_2$ simulations show a small overestimation with a mean bias error of 2.52 ppm. For CH$_4$ we find a model underestimation of -3.03 ppb for the columns and an overestimation of 9.43 ppb near the surface. Finally, we find no significant

bias for the XCO time series (0.42 ppb). The bias is below the measurement uncertainty only for XCH$_4$ and XCO. Furthermore, relatively high correlation coefficients ($\geq 0.68$) are found for all CO$_2$ and CO time series. For CH$_4$ only a moderate correlation of 0.56 was found.

In Fig. 3a, we see that WRF-GHG is underestimating XCO$_2$ until May 2019 with about 2 ppm, after which the negative bias disappears. A similar pattern is found when comparing the CAMS reanalysis data set with the TCCON data at Xianghe and

other sites in that part of the globe (Rikubetsu, Tsukuba, Saga, Hefei). As CAMS reanalysis data was used as initial and lateral boundary conditions, we assume that the error pattern detected in the XCO$_2$ time series is the result of the same pattern in the background information. Moreover, this bias is not found in the in situ CO$_2$ time series (Fig. 3b), likely because the relative contributions from emission sources in the domain to the in situ concentrations are much larger than they are to the column data.

Similarly, it can be seen from Fig. 3c that WRF-GHG is underestimating the TCCON XCH$_4$ data in summer (June - November) and slightly overestimating them in winter (January - March), leading to the lower correlation coefficient. Again, a comparable pattern is found in the CAMS - TCCON comparison, however with a much smaller amplitude. Moreover, the same seasonal bias is found in the time series for the in situ data (Fig. 3d). Therefore, this discrepancy is likely linked to an incorrect seasonality in the CH$_4$ emissions. More detail on possible sources of this mismatch will be given in Sect. 6.3.

**6 Sector contributions**

As explained in Sect. 3, all fluxes that are included in WRF-GHG are tracked in separate tracers. This allows us to disentangle the total simulated concentrations into the different tracer contributions and evaluate the influence of different source sectors





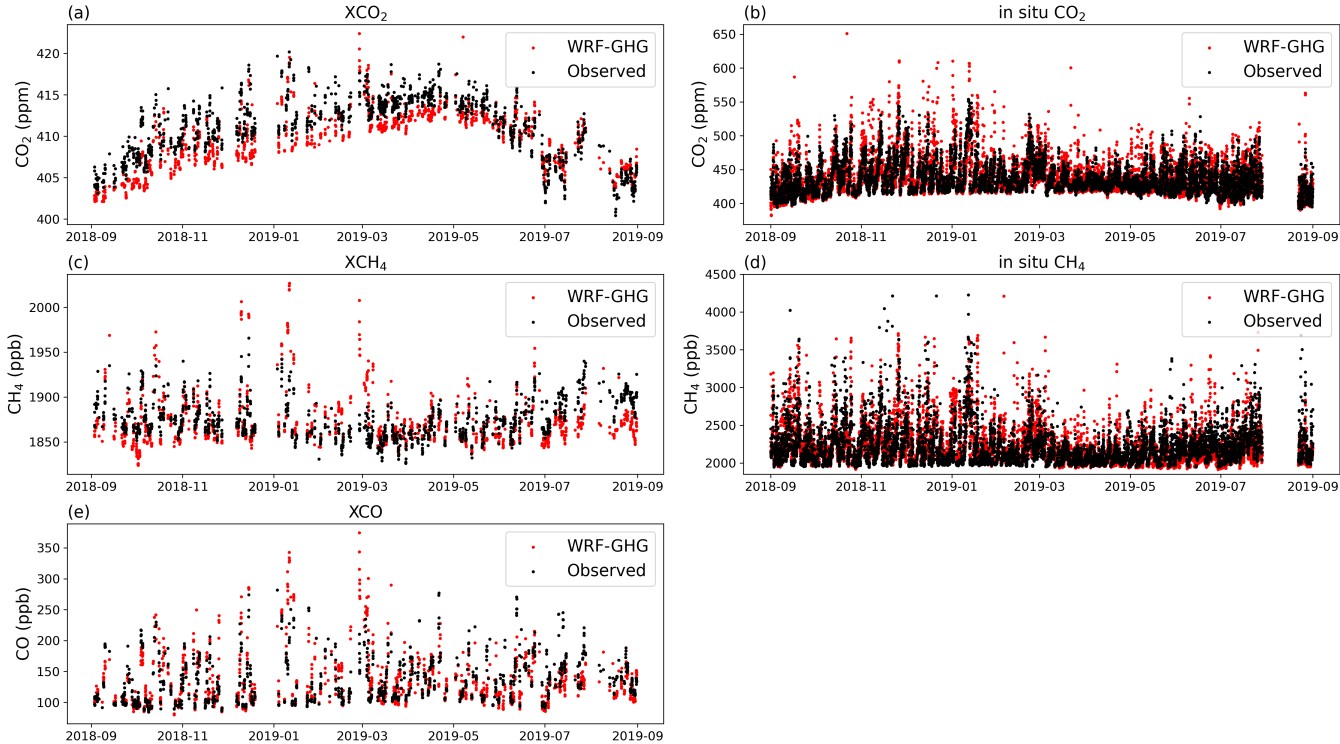

**Figure 2.** Observed (black) and simulated (red) time series at the Xianghe site of (a) XCO$_2$, (b) in situ CO$_2$, (c) XCH$_4$, (d) in situ CH$_4$ and (e) XCO. Data points are hourly.

on the observations at Xianghe, as well as their respective importance. An overview of the monthly mean values is shown in Fig. 4. Note that all simulated hours were used for this plot, not just the ones coinciding with observations.

## 6.1 CO

According to WRF-GHG, the CO column time series is primarily influenced by sources from the industry, residential and transportation sectors, of which the industry sector is the largest contributor (Fig. 4e). Energy sources and biomass burning are not important for the observations at Xianghe. Both residential and transportation tracers show larger values in winter, which is in agreement with higher emissions in that period of the year due to colder air temperatures. Remark that the emissions from the industry sectors are relatively constant throughout the year, while the tracer contributions to the simulated concentrations in Xianghe show quite some variability with for example larger values in winter and smaller in spring. This is likely because of the variability in meteorological conditions (wind direction, stagnant air masses etc ...) from month to month, which we will come back to in Sect. 8.



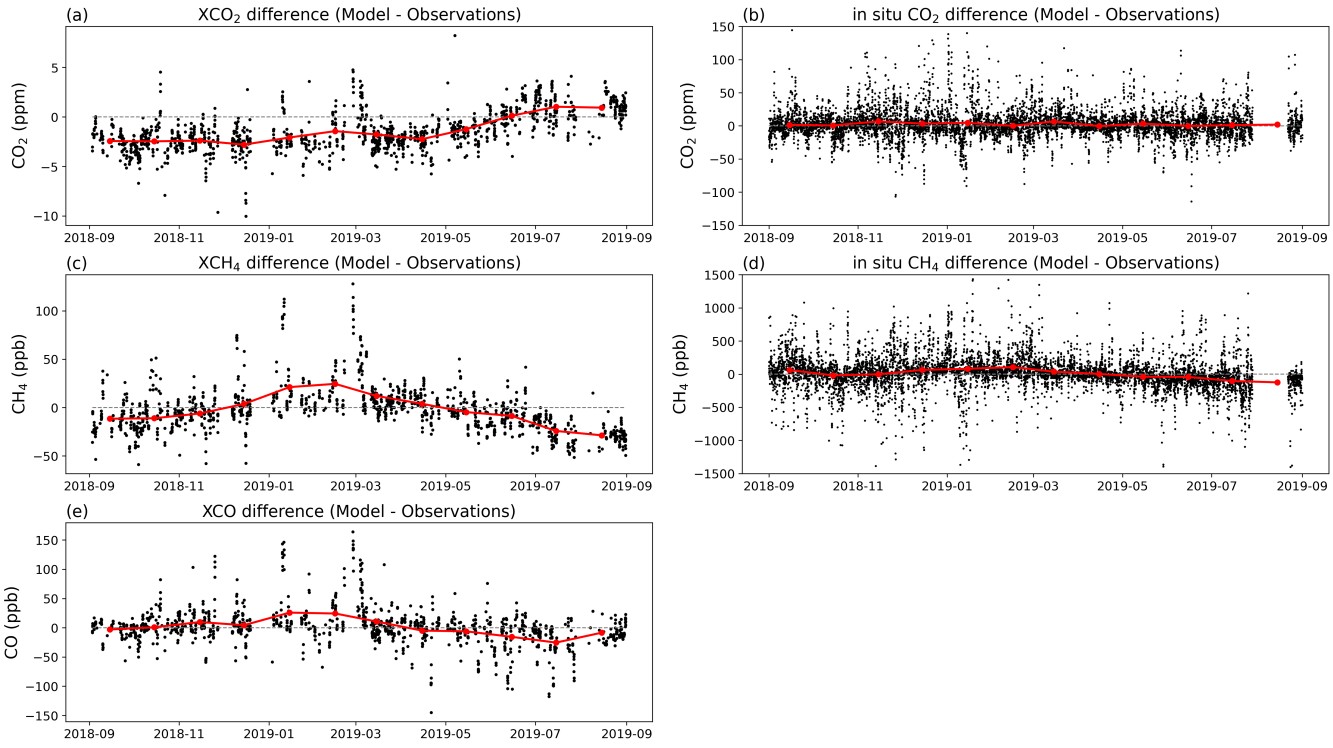

**Figure 3.** Time series of the differences between WRF-GHG simulations and observations of (a) XCO$_2$, (b) in situ CO$_2$, (c) XCH$_4$, (d) in situ CH$_4$ and (e) XCO. The black dots are hourly values, while the red line indicates the monthly mean differences.

## 6.2  CO$_2$

The main sectors contributing to the CO$_2$ data at Xianghe are energy, industry and the biosphere (Fig. 4a,b). For both column and surface simulations, the largest sectors are energy (which is mainly power plants) and industry. Additionally, the biosphere contributes significantly to the column data, especially from May to September. In the rest of the year, the biogenic tracer is a net source. For the surface data the biosphere is a small source throughout the year, except in August. The difference between column and in situ can be attributed to the fact that the regions with the most pronounced biogenic sink are a bit

further away from Xianghe and are therefore better sensed by the column observations than those near the surface. Likewise, the VPRM-computed fluxes indicate that the biosphere in the immediate vicinity of Xianghe predominantly acts a net source, with the exception of the month of August. Next to the biosphere, industry and energy, also transportation and residential sources have a small influence on the Xianghe data. During winter, the contribution of residential sources is larger than in the rest of the year, which is in agreement with the general emissions patterns in China. Finally, no relevant impact was found

from biomass burning and the ocean. As for CO, the tracer contributions from the energy and industry sectors show quite some variability despite their relatively constant emissions throughout the year. This can likely be attributed to meteorological factors as discussed further in a later section.





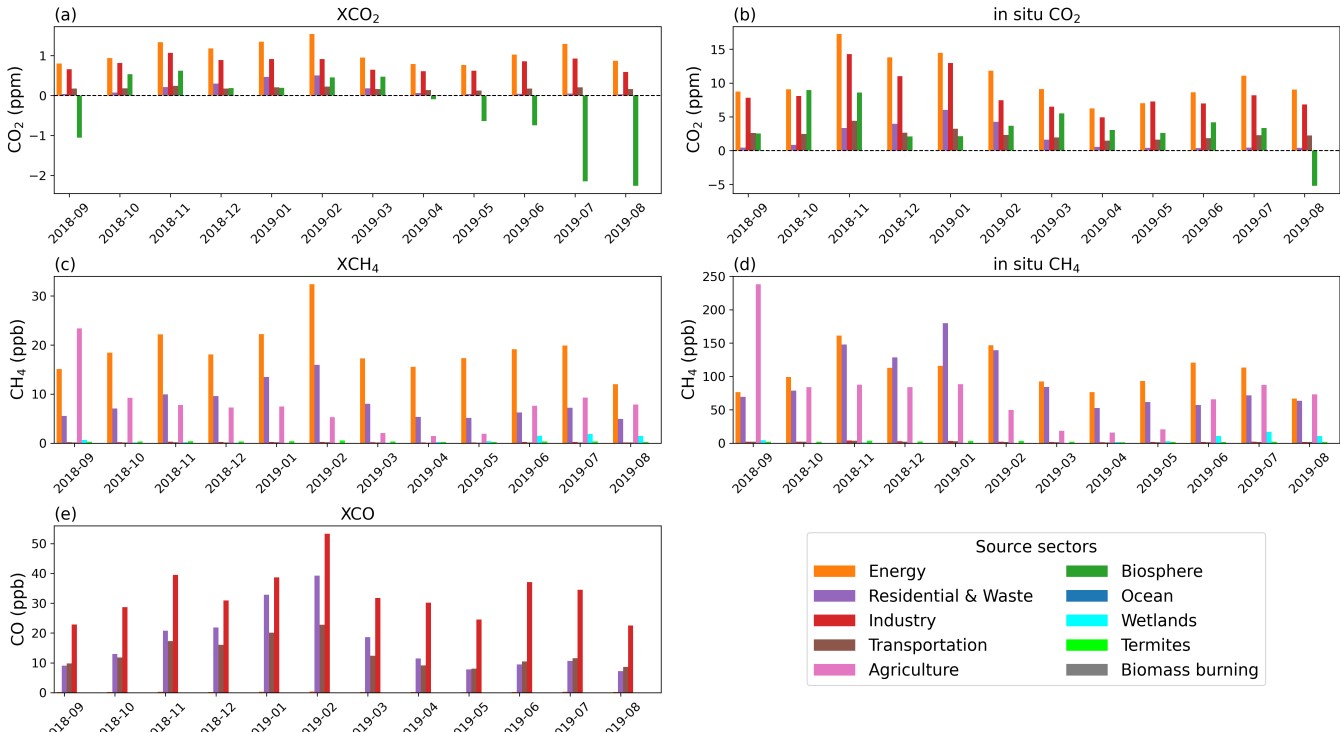

**Figure 4.** Monthly mean tracer contributions above the background for (a) $XCO_2$, (b) in situ $CO_2$, (c) $XCH_4$, (d) in situ $CH_4$ and (e) XCO simulated concentrations at Xianghe.

### 6.3  $CH_4$

For $CH_4$ the simulated signal at Xianghe is mainly influenced by three sectors: energy, agriculture and residential & waste

(Fig. 4c,d). Furthermore there is a small contribution from wetlands in summer. Other sectors such as industry, transportation, termites and biomass burning seem to be irrelevant at Xianghe. Energy sources appear to have more impact on the column than on the in situ observations. When looking at the mean vertical profiles of the different tracer contributions above Xianghe (Fig. B1) we see that the contributions from the energy sector are generally found at a higher altitude compared to other sectors. High concentrations near the surface are associated with emission sources nearby, while those aloft are likely caused

by long-distance pollutant transport in the free troposphere. Therefore, we assume that this difference between column and surface energy contribution is because the strongest energy sources are situated in Shanxi (the largest coal producing province in China), which is much further away from Xianghe than for example the strongest residential sources (Beijing and Tianjin), see Fig. B2.

There is a larger residential signal in winter, while for agriculture, the contributions peak in September and are smallest in

spring. This corresponds with the seasonal pattern of emissions within CAMS-GLOB-ANT.

As for the other species, we see quite some month-to-month variability in the contribution of the energy sector, which is



probably due to differences in meteorological conditions as the month-to-month variations are quite similar for the different tracers and the other gases, despite their different sources.

## 7  Seasonal CH$_4$ bias

In Sect. 5 we discussed that there is a seasonal bias in the CH$_4$ time series of WRF-GHG which is likely caused by incorrect emission fluxes. Assuming that the main sectors as simulated by WRF-GHG and discussed in Sect. 6.3 are realistic, we suspect the following sources of error:

– Agriculture. As presented in Table 3, the agricultural sector is comprised of three subsectors: soils (this is mainly rice cultivation), agricultural waste burning, and livestock (manure management and enteric fermentation). In China, rice
cultivation plays a vital role but is predominantly concentrated in regions south of 35°N. According to CAMS-GLOB-ANT, the most important agriculture subsector in the region of the Xianghe site is livestock. Unfortunately, the source of monthly variations in CH$_4$ emissions within the CAMS-GLOB-ANT data set remains somewhat unclear, as the accompanying data set of temporal factors, CAMS-GLOB-TEMPO (Guevara et al., 2021), references constant factors for CH$_4$ emissions from agricultural sources. The data set however, shows peak emissions in September in the wide region
around Xianghe and minimum values in March and April. Maasakkers et al. (2016) suggests that manure management is dependent on air temperature, which would lead to larger emissions from May to September and lower from December to February. If this theory holds, it implies we would find an underestimation of CH$_4$ concentrations in the summer and an overestimation in the winter. Therefore, an incorrect monthly variation in the CH$_4$ agricultural emissions could potentially account for the observed bias.

– Residential & waste. This sector represents emissions from residential, commercial and other combustion sources together with CH$_4$ emissions from solid waste and waste water treatment. In CAMS-GLOB-ANT, the waste sector is the most important one in the Xianghe region and assumed to be constant throughout the year. In winter, when residential combustion emissions are higher, they are almost similar in size as the waste emissions. So the seasonality of the residential & waste sector is coming from the residential part, peaking in winter. This is consistent with the simulated
contributions from this sector (see Fig. 4c,d). However, Hu et al. (2023) showed that CH$_4$ emissions from waste treatment often follow the seasonality of air temperature. Even though this study is based on observations in the Hangzhou megacity, their results could possibly be representative for the BTH region as well. This would mean that the waste emissions are underestimated in summer and/or overestimated in winter, which would match the current model-observation mismatch for CH$_4$.

– Wetlands. Within the WRF-GHG simulations, wetlands only show minor contributions to the surface and column data, and only in summer. Emissions are taken from the WetCHARTs v1.0 ensemble data set. In the BTH area, the main wetland areas are located close to the Bohai Sea (see Fig. B2). However, according to WetCHARTs, these emissions are relatively small compared to those from wetlands more in the south of China. In an evaluation of the WetCHARTs





ensemble against GOSAT observations by Parker et al. (2020), a general underestimation of the seasonal amplitude in
China was found. This would mean an underestimation of the wetland $CH_4$ emissions in summer. Furthermore, Chen
et al. (2022) showed increased posterior wetlands emissions compared to the a priori values when inferring yearly $CH_4$
emissions over China using TROPOMI satellite observations. This could point to an underestimation of the wetland
emissions in the current study.

The observed seasonal error pattern between the WRF-GHG $CH_4$ simulations and the observations at Xianghe could be caused
by one of the reasons mentioned above, or by a combination of them. For a more spatial perspective of this seasonal bias, we
compared the WRF-GHG $XCH_4$ field with TROPOMI observations. Figure 5 shows the seasonal mean $XCH_4$ from WRF-
GHG and TROPOMI over the wide region around Xianghe. In DJF and MAM both maps agree quite well, however in JJA and
SON there are large differences between WRF-GHG and TROPOMI. WRF-GHG seems to underestimate $CH_4$ from June to
November and this over the entire area. This corresponds with our previous findings in Sect. 5 when comparing WRF-GHG
with TCCON and confirms the good agreement between TROPOMI and TCCON in this region. The largest discrepancies are
found in JJA, when there are much higher concentrations measured by TROPOMI, especially to the south of 38°N.

Because Fig. 5 shows differences on a large spatial scale, this indicates that the underestimation by WRF-GHG is linked to
emission sources that are widespread in the region. Since the North China Plain is a livestock-dominated region with strong
urbanization and industrial activities we assume that it is the fluxes of either agriculture (livestock), waste treatment or both,
rather than the fluxes from wetlands, that are underestimated in summer in CAMS-GLOB-ANT.

More research about the seasonality of $CH_4$ emissions in north China is needed to understand these discrepancies.

## 8 Meteorological factors influencing variability

The previous sections discussed how emissions from different sources affect the $CO_2$, $CH_4$ and CO observations at Xianghe.
In the current section we want to focus on the meteorological factors that influence the temporal variability of the time series.
More specifically we will discuss the impact of large-scale phenomena, the planetary boundary layer and local winds.

### 8.1 Synoptic scale winds

Because FTIR observations generally have a large area of representativeness (generally a few 100 km), column concentrations
are relatively insensitive to local fluxes and vertical mixing, while they are strongly influenced by large-scale patterns (Keppel-
Aleks et al., 2011). We use the winds at 800 hPa to represent horizontal transport in the free troposphere, as this altitude is
generally above the planetary boundary layer height. More specifically, we looked at the daily mean column concentrations
above the background for every wind direction to see if a clear relationship could be found. This is shown in Fig. 6.

Remark that only southwest (SW) and northwest (NW) wind segments are given because southeast and northeast winds occur
only seldom at 800 hPa: only on 2 and 15 days out of 237, respectively. We find that in general, larger enhancements are
found when winds blow from the SW wind segment compared to the NW segment. To quantify the difference, we performed a
non-parametric Mann-Whitney U test on the two categories. For all species, we find p-values far below 0.05 (see Fig. 6, on top





**Figure 5.** Seasonal mean XCH$_4$ (ppb) over the domain d02 (provinces of Beijing, Tianjin, Hebei, Shanxi and part of Shandong) as simulated by WRF-GHG (first column) and observed by TROPOMI (second column). The seasons are defined as (a,b) SON: September - November (autumn), (c,d) DJF: December - February (winter), (e,f) MAM: March - April (spring) and (g,h) JJA: June - August (summer). White pixels indicate that there are no observations available during the entire period.



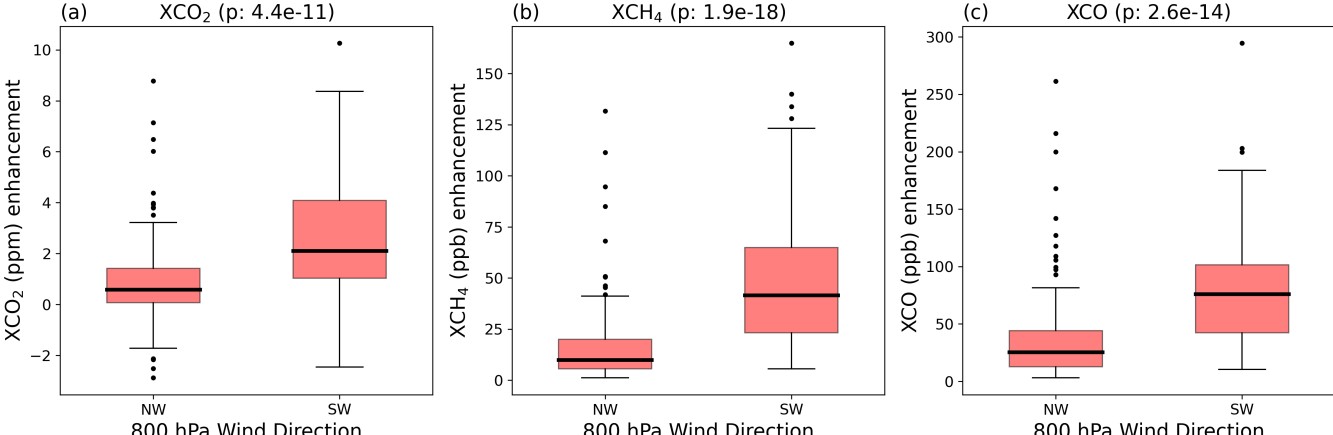

**Figure 6.** The distribution of the daily mean simulated column tracers above the background per 800 hPa wind direction category and species. NW is for winds with an angle of 270 to 360 ° from north, while SW represents the angles between 180 and 270°. There are 134 days with NW winds and 86 days with SW winds. The colored boxes indicate the range between the first and third quartile, while the thick solid line is the median. Outliers (values that are 1.5 times the interquartile range above (below) the third (first) quartile) are shown by black dots.

of each panel) indicating that indeed the differences are statistically significant. Higher concentrations coincide with 800 hPa winds coming from the SW while NW winds correspond with lower concentrations. Yang et al. (2020) already showed that the day-to-day variation of the column observations of the different species are highly intercorrelated, and that clean days are linked with air from the north, while polluted days are linked with air from the south, which is confirmed here by the WRF-
GHG simulations. Air masses from the north have been moving over rather remote and clean areas such as Inner Mongolia, Mongolia and Russia. Meanwhile, southerly air is linked with the highly populated North China Plain (NCP), where many emission sources are located.

As discussed in Sect. 6, absolute tracer contributions are also affected by the monthly variability in meteorological phenomena. For example in the winter months, weather conditions are generally more favorable to accumulation leading to high pollution
levels (Li et al., 2022). This could enhance both local plumes as well as those advected with the southwestern winds. The tracer concentrations at Xianghe are therefore the result of a complex interaction of local and remote emissions, wind direction and local and remote weather patterns.

Moreover, the presence of polluted air from the SW is also visible in the surface concentrations. A high correlation coefficient of 0.79 is found between the daily mean column and surface tracer contributions above the background for both $CO_2$ and $CH_4$
(see Fig. 7). This means that both the surface and column concentrations are influenced by synoptic scale winds, bringing either clean or polluted air masses to Xianghe.





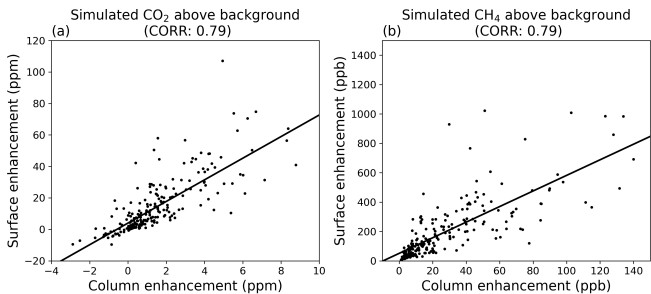

**Figure 7.** Daily mean surface enhancement with respect to column enhancement for (a) $CO_2$ and (b) $CH_4$. The enhancement is defined as the difference between the total simulated concentration and the background tracer concentration.

## 8.2 Planetary boundary layer dynamics

The planetary boundary layer (PBL) is the lowermost layer of the atmosphere which is in direct contact with the Earth's surface. The characteristics of this layer vary throughout the day. During the day, under influence of solar radiation, turbulent motions cause strong vertical mixing of the air within the PBL. These processes allow gases to be dispersed and transported upwards, which generally leads to reduced concentrations near the surface. At night, radiational cooling of the surface creates a temperature inversion close to the ground. This stable nocturnal layer is quite shallow and tends to trap pollutants near the surface, increasing their concentrations.

Figure 8 shows the diurnal variation of the PBL height as simulated by WRF-GHG and the $CO_2$ and $CH_4$ concentrations near the surface (both simulated and observed). Indeed, the height of the PBL is largest in the afternoon when solar radiation is strongest, reaching its peak at 15h (local time). This corresponds with the lowest observed surface concentrations (Fig. 8b,c). Right after sunset, the height of the PBL drops to its lowest value, after which it persists during the course of the night, until sunrise. This period corresponds with slightly increasing $CO_2$ and $CH_4$ concentrations as emissions near the surface accumulate within this stable shallow layer. The highest concentrations are found in the early morning, around 7-9h local time, when the PBL is still quite shallow and the emissions are peaking. Afterwards, when turbulent mixing has emerged, the concentrations suddenly drop, creating a diurnal cycle. Remark that WRF-GHG is very well capable at simulating this diurnal variation of both $CO_2$ and $CH_4$ in situ observations. These dynamics are very important for the variability of the surface concentrations, however they are irrelevant for the column concentrations, as the latter are much less affected by vertical transport (Wunch et al., 2011). Moreover, the FTIR observations are only available during the day because they require the presence of solar radiation.

## 8.3 Local emissions

Regional emissions are influencing both column and in situ concentrations at Xianghe, as elaborated in Sect. 8.1. However, emission sources nearby could also have an impact on these values, especially for the in situ observations as they sample the





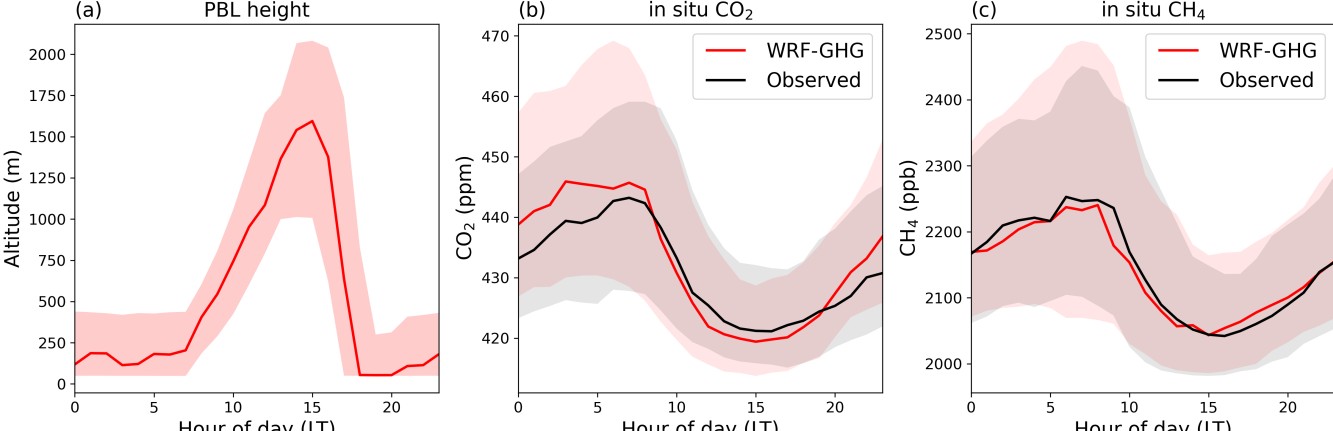

**Figure 8.** Hourly median and interquartile range of the (a) simulated planetary boundary layer height, and observed and simulated surface (b) $CO_2$ and (c) $CH_4$ concentration at Xianghe.

local air. To analyze which nearby sources influence the Xianghe measurements, we look for correlations between the 10m wind speed and direction and the simulated concentrations. Figure 9 reveals the mean WRF-GHG tracer contribution per wind direction and speed for $CH_4$. To eliminate the influence of polluted plumes from further away, we select only those days on which the mean daily XCO enhancement (sum of all WRF-GHG tracers above the background) is smaller than 45 ppb. We use XCO as a tracer for polluted events as it is the species with the shortest atmospheric lifetime. Furthermore, we compute the mean concentrations separately for day and night to avoid the effects of the PBL. The night hours are defined as those with the peak concentrations, i.e., between 3h and 8h LT, while the day represents those hours with highest atmospheric mixing and lowest concentrations, i.e., between 13h and 18h. During the day most winds are coming from the north and southwest, while at night the most frequent wind directions are north and east. Higher wind speeds are found during the day than at night. The northern winds typically have the lowest tracer contributions since there are fewer emission sources in this direction, with the exception of agriculture (see Fig. 10). In general, we see that wind directions with the largest enhancements correspond with the largest sources nearby (Fig. 9-10): east and west for energy, all but north for residential, all directions for agriculture, southwest for industry and southeast for wetlands. Similar plots for $CO_2$ are given in Fig. B3 and B4. The highest values overall are found for the energy tracer at night and they are coming from the east. To the east are some very large $CH_4$ point sources which correspond to coal mine emissions nearby the city of Tangshan (see Fig. 10a). However, when looking closer at the $CH_4$ time series we see that WRF-GHG is often overestimating the Xianghe in situ $CH_4$ observations at times where the model shows a large energy contribution. This is also visible in Fig. 11. This makes us to believe that these coal mine emissions might be overestimated in CAMS-GLOB-ANT. In the next section we try to verify this assumption by comparing WRF-GHG concentration fields with TROPOMI observations.




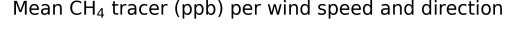

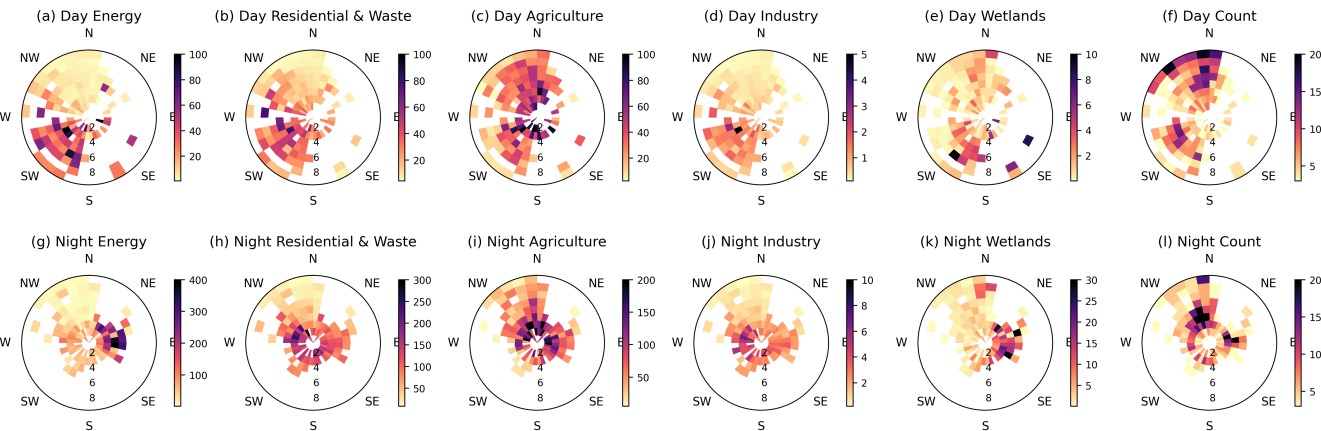

**Figure 9.** Mean CH$_4$ simulated tracer concentrations (indicated by colour scale, in ppb) binned per wind speed and direction for the main sectors (a) energy, (b) residential & waste, (c) agriculture, (d) industry and (f) wetlands on days without strong regional pollution. The first row represents afternoon hours (13h - 18h LT), while the second row represents nighttime hours (3h - 8h LT). Data is binned per 1 m s$^{-1}$ and 11.25° wind direction. (f) Count of data points in each bin. Only bins with at least 3 points are included in the figure. Remark that the panels have different colour scales.

### 8.3.1 Assessing CH$_4$ emission sources

By comparing the TROPOMI XCH$_4$ with WRF-GHG XCH$_4$, we want to assess if the CH$_4$ emissions from coal mines around Tangshan are indeed overestimated in CAMS-GLOB-ANT or not. Figure 12 shows the maps of the weighted mean XCH$_4$ during the entire simulation period: September 2018 until September 2019. The yearly mean total CH$_4$ fluxes from CAMS-GLOB-ANT in the WRF-GHG domain 2 is also given. By taking the average over the complete simulation period we aim to minimize the influence of meteorological patterns on the XCH$_4$ concentration and expose the main emission sources. When comparing the WRF-GHG input fluxes in Fig. 12a with the resulting XCH$_4$ concentration field in Fig. 12b, we indeed find a strong agreement. The largest sources are found to the west of 114°E, which correspond to the extensive coal mining activities in Shanxi. In the same locations on the XCH$_4$ map we find the highest concentration values of the region. Unfortunately due to the mountainous terrain, TROPOMI observations are sparse in this area. Other sources, such as a hotspot around 36°N, 117°E and the slightly smaller emissions around Beijing (40°N, 116.3°E) and Tangshan (39.6°N, 118.4°E) correspond with elevated XCH$_4$ values. This suggests that yearly averaged XCH$_4$ maps can indeed reveal the strongest emission sources. The region below 37°N shows high simulated XCH$_4$ values as well, however they do not directly correspond to strong sources in the inventory. This can likely be explained by the presence of the Taihang mountains on the west which lead to poor dispersion conditions (Fu et al., 2014). Therefore the larger concentrations in this area are likely more determined by the topography and associated meteorological conditions than by surface fluxes.

We find slightly elevated XCH$_4$ values nearby the coal mines of Tangshan ($\sim$ 39.6°N, 118.4°E) in both WRF-GHG and




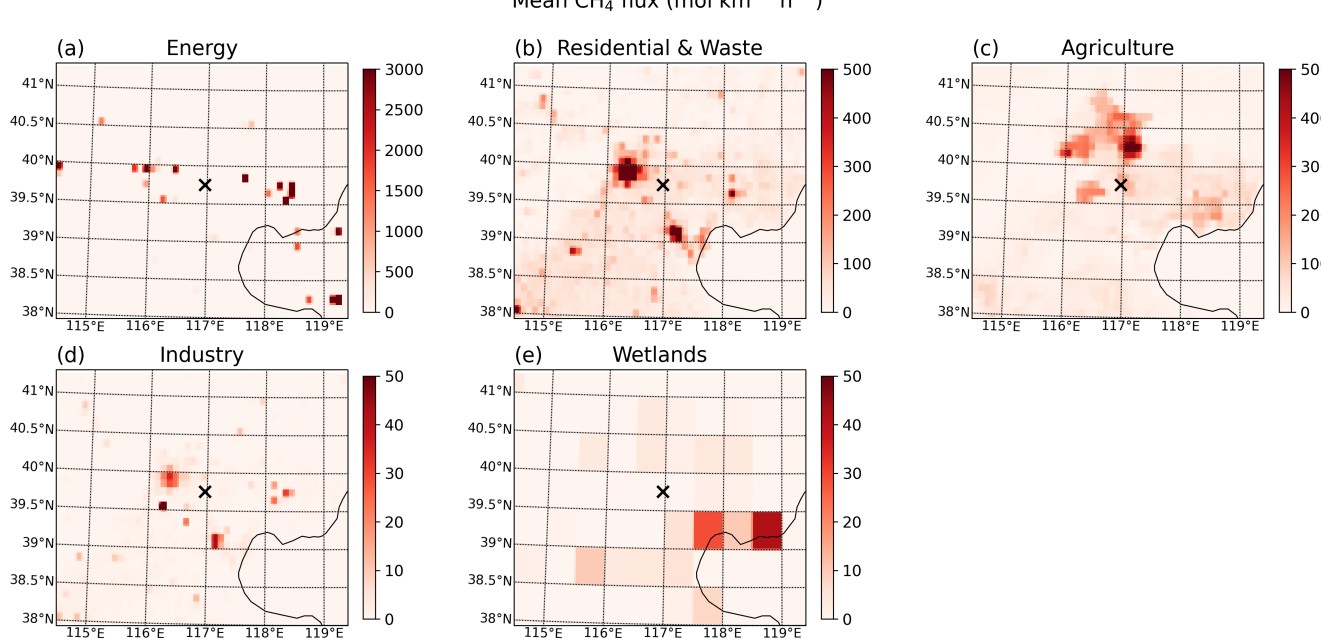

**Figure 10.** Map of the mean CH$_4$ flux (mol km$^{-2}$ h$^{-1}$) in WRF-GHG domain d03 during the entire simulation period from September 2018 until September 2019, for the most important sectors. Remark that the panels have different colour bar scale. The location of the Xianghe site is indicated by a black cross.

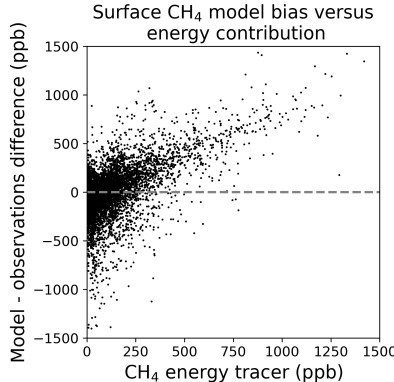

**Figure 11.** Correlation between energy tracer contribution to simulated CH$_4$ surface concentrations and differences between total simulated and observed surface concentrations. For this plot, the data was not filtered on day, night or polluted/clean days.



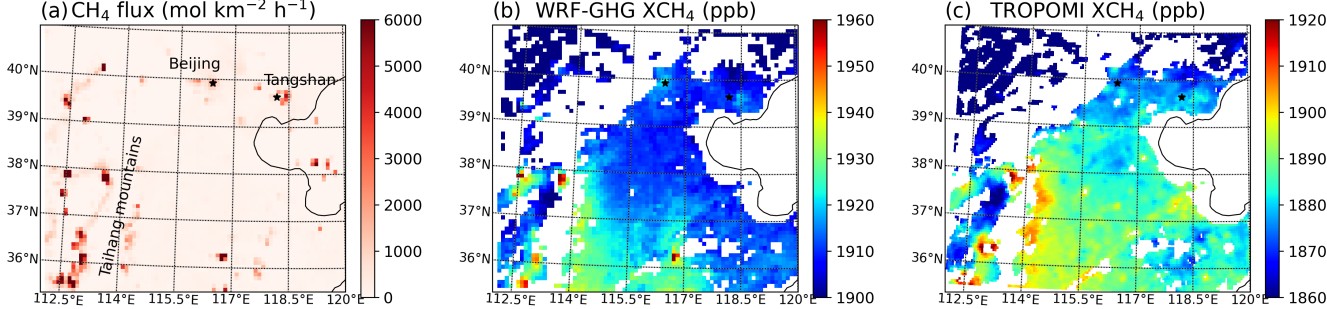

**Figure 12.** (a) The $CH_4$ flux from all sectors in CAMS-GLOB-ANT averaged from September 2018 until September 2019 and regrid to WRF-GHG grid d02 (9 km resolution). Mean $XCH_4$ over the same period as (b) simulated by WRF-GHG in domain d02 and (c) observed by TROPOMI (regridded to 0.1 °).

TROPOMI maps. However, the TROPOMI map shows similar or even stronger elevated concentrations at many other locations in the region, indicating a general $XCH_4$ underestimation by WRF-GHG. Remind that we applied a fixed offset of 40 ppb to the colour scale of the WRF-GHG plot to account for the $CH_4$ concentrations above 50 hPa, this is however an approximation so comparing both maps in a quantitative way should be avoided. If we consider only the colour gradients instead, we can deduce that the $CH_4$ sources around Tangshan (and Beijing) are likely overestimated *relative to* others in the region. To put it differently,

it is possible that either most sources within BTH are accurately represented in the WRF-GHG model, except for those in the vicinity of Beijing and Tangshan, which are overestimated. Alternatively, it could be the reverse scenario, where emissions from Beijing and Tangshan are relatively accurate, but those from the other sources in the region are underestimated. Nevertheless, it is important to note that this qualitative comparison alone cannot yield a definitive conclusion since the considered sources are likely too small to be reliably detected by TROPOMI.

**9 Conclusions**

We have used the WRF-Chem model in its passive tracer option WRF-GHG to simulate surface concentrations and column abundances of $CO_2$, $CH_4$ and CO observed at the Xianghe site in China and to improve our knowledge about the origin of the observed variabilities in the measured time series. Since June 2018, column-averaged concentrations are measured with a FTIR spectrometer that is part of TCCON, while near-surface concentrations of $CO_2$ and $CH_4$ are measured with a PICARRO

CRDS analyzer at an altitude of 60 m.a.g.l. With WRF-GHG we computed 3-D concentration fields from September 2018 until September 2019 in three nested domains covering a large part of China and its neighboring countries. The simulations from the innermost domain covering the Beijing-Tianjin-Hebei megalopolis with a horizontal resolution of $3\times3$ km$^2$ are compared with the observations. Sensitivity tests were performed to select the most suitable anthropogenic emission inventory and set of physical parameterization schemes. The CAMS-GLOB-ANT v5.3 inventory was selected to represent anthropogenic

emissions within the model domain for $CO_2$ and $CH_4$, while for CO we have used the REAS v3.2.1 data set. For all species,



different subsectors were aggregated into five source sectors to analyze their respective contributions: industry, energy, agriculture, transportation and residential & waste. Other sources that were taken into account were biomass burning, wetlands, termites, and $CO_2$ fluxes from ocean and the biosphere.

Overall, we found a good model performance with correlation coefficients above 0.68 for $CO_2$ and CO and 0.56 for $CH_4$. A negligible bias was found for XCO while WRF-GHG showed a small underestimation of -1.43 ppm and -3.03 ppb with respect to TCCON $XCO_2$ and $XCH_4$, respectively. The in situ time series of $CO_2$ and $CH_4$ are slightly overestimated by WRF-GHG, by 2.52 ppm and 9.43 ppb, respectively.

For $CO_2$, the most important sectors contributing to the Xianghe observations are industry, energy and the biosphere, followed by minor contributions from residential and transportation sources. The $CH_4$ signal mainly consists of enhancements from energy, agriculture and residential & waste sectors. And finally, for CO we found that industry is the largest emission sector followed by residential and transportation. For $CO_2$, the biogenic contributions to the column observations are larger in summer when they are a sink, while they are a net source in winter. Near the surface, the relative biogenic $CO_2$ contributions are smaller and a source in all months except August. This difference is likely because of the lack of strong photo-synthetically active vegetation in the neighborhood of Xianghe. For $CH_4$, the energy sector appears to have a larger impact on the column abundances than on the surface concentrations due to differences in the sensitivity footprints between the remote sensing column and in situ observation and the distance between the observation site and the strongest sources. For all species, the residential and transportation sectors show larger contributions in winter compared to summer, consistent with their emission patterns. However, all tracers are additionally influenced by monthly variability in meteorological conditions such as horizontal advection and atmospheric stability. This is especially visible in the contributions of energy and industry sectors which have relatively constant emissions throughout the year.

A strong correlation between the column enhancements and the free tropospheric wind direction showed that air masses advected from the southwest generally carry higher concentrations compared to those from the northwest. Under southwest wind regimes, pollution from the heavily populated and industrialized North China Plain reaches the Xianghe site. This increases both the column abundances and surface concentrations. Due to their large spatial footprint, these large-scale phenomena are the dominant factor influencing the variability of the column data. However for the in situ observations, also the planetary boundary layer and local emissions play an important role. The daily cycle of turbulent mixing during the day and accumulation near the surface at night driven by solar radiation leads to lowest $CO_2$ and $CH_4$ concentrations being measured in the afternoon when atmospheric mixing is strongest. On the other hand, peak values are found in the early morning around sunrise when local emissions are strongest and species have been accumulated all night in the lowest layer of the atmosphere. Depending on the local wind speed and direction, plumes from nearby sources are visible in the Xianghe observations.

Some discrepancies in the model-data comparisons are difficult to explain with WRF-GHG and local observations at Xianghe alone. Therefore, we additionally compared the simulated $XCH_4$ with observations from TROPOMI to support our initial findings. More specifically we found that WRF-GHG is underestimating $CH_4$ at Xianghe both near the surface and in the total column from June to September. Part of this seasonal bias can be explained by a similar bias that is already present in the lateral boundary conditions from the CAMS reanalysis. Additionally we suspect that widespread emission sources such as





agriculture and waste have incorrect seasonal cycles in the CAMS-GLOB-ANT inventory as a similar difference pattern was found between WRF-GHG and TROPOMI, but on a larger scale. Secondly, comparisons between the simulated and observed $CH_4$ concentrations near the surface identified a possible overestimation of the $CH_4$ emissions around Tangshan which are associated with coal mining. Because column-averaged concentrations are integrated over multiple atmospheric layers, only

very strong point sources are easily detectable by TROPOMI $XCH_4$ observations and the energy sources around Tangshan lead to only minor enhancements in the TROPOMI $XCH_4$ fields. Therefore the TROPOMI observations can not confirm the overestimation of the $CH_4$ coal mine emissions, nor do they contradict this assumption.

This study showed that high-resolution simulations of WRF-GHG are useful to analyze both remote sensing and in situ observations of $CO_2$, $CH_4$ and CO column abundances and surface concentrations at Xianghe. The added value of the source sector

separation and the simultaneous calculated meteorological fields yields an extensive data set, enabling us to give a variety of new perspectives on the observed time series. The simulated fields, however, strongly rely on accurate boundary conditions because the considered species have relatively long atmospheric lifetimes, and inaccuracies therein will be transmitted into the model's output. The dependence on precise emission data can be both a strength and a weakness, as discrepancies between simulated and observed fields can be used to identify flaws in bottom-up emission inventories.

*Code and data availability.*    The ERA5 and CAMS reanalysis data set (Hersbach et al., 2023a, b), used as input for the WRF-GHG simulations, was downloaded from the Copernicus Climate Change Service (C3S) Climate Data Store (2022). The CAMS-GLOB-ANT v5.3 emissions (Granier et al., 2019; Soulie et al., 2023) and temporal profiles CAMS-GLOB-TEMPO v3.1 (Guevara et al., 2021) are archived and distributed through the Emissions of atmospheric Compounds and Compilation of Ancillary Data (ECCAD) platform. The REAS emission inventory is publicly available at https://www.nies.go.jp/REAS/ (Kurokawa and Ohara, 2020). The WRF-Chem model code is distributed by

NCAR (https://doi.org/10.5065/D6MK6B4K, NCAR, 2020). The WRF-GHG simulation output created in the context of this study can be accessed on https://doi.org/10.18758/P34WJEW2 (Callewaert, 2023). The TCCON data were obtained from the TCCON Data Archive hosted by CaltechDATA at https://tccondata.org (Zhou et al., 2022), while the surface observations at Xianghe were received through private communication with the co-authors. TROPOMI Level 2 Methane Total Column data are publicly available online at https://doi.org/10.5270/S5P-3lcdqiv and the Copernicus Open Access Hub.

**Appendix A: WRF-GHG sensitivity tests**

In Sect. 3.2 we explained that different anthropogenic emission inventories and physical parameterization schemes were tested in the WRF-GHG simulations. More specifically, up to 7 different flux data sets (depending on the species) and 5 different parameterization configurations were considered (see Table 1). The model output of simulations over three short periods was combined and compared with the observational data based on the statistical metrics of root mean square error (RMSE), mean

bias error (BIAS) and Pearson correlation coefficient (CORR). An overview of all results, together with the combined skill score ($S$, Eq. 1) is given in Tables A1, A2, A3, A4 and A5 for the time series of in situ $CO_2$, in situ $CH_4$, $XCO_2$, $XCH_4$ and





XCO, respectively.

As the combination with the highest $S$ is different among the five observation types, we decided on the final set as follows:


– For each statistical metric, we calculate a threshold derived from the mean ($\mu$) and standard deviation ($\sigma$) of all occurring values. Combinations in which one or more of the metrics exceed or fall below these thresholds are excluded from the selection process. Specifically, these combinations must conform to the following set of equations:

$$\text{CORR} \geq \mu_{\text{CORR}} - \sigma_{\text{CORR}},$$
$$|\text{BIAS}| \leq \mu_{|\text{BIAS}|} + \sigma_{|\text{BIAS}|},$$
$$\text{RMSE} \leq \mu_{\text{RMSE}} + \sigma_{\text{RMSE}} \tag{A1}$$

The combinations that are discarded after this step are highlighted in dark gray in the tables below.


– For $CO_2$ and $CH_4$, discard the combinations that are only present in the table of either the surface or either the column data in order to keep only those that are performing good enough on both time series. The combinations that are discarded after this step are highlighted in light gray in the tables below.

– From what is left, we see that only combinations with test A, B or C should be considered as those with test D and E settings have been discarded for $CH_4$. The choice of physical parameterization option should be the same for all species. When sorting the remaining combinations for $CO_2$ and CO based on $S$ (from the in situ time series for $CO_2$), we find


that options with test B and C are superior to those with test A. Finally, a choice has to be made between options with test B and options with test C.

– For both test B and C, we take the emission inventory which has the highest $S$, for $CO_2$ and $CH_4$ based on the in situ time series and for CO based on the column. This leads to the following options:

– Test B: CAMS-GLOB-ANT for $CH_4$ and $CO_2$; REAS for CO


– Test C: CAMS-GLOB-ANT for $CH_4$, REAS for $CO_2$ and PKU for CO

– The final choice between these two options is rather arbitrary since certain combinations yield slightly improved results for one time series but perform less favorably for another, and vice versa. In our study we have chose the combinations with test B.

## Appendix B: Supplementary figures

This appendix contains figures that give some additional insight to the conclusions given in the sections above and are referenced in the text.



| Test | Flux | CORR | BIAS | RMSE | S |
|------|------|------|------|------|------|
| B | PKU | 0.67 | -1.62 | 16.09 | 2.91 |
| **B** | **CAMS** | **0.63** | **-0.12** | **17.50** | **2.81** |
| B | EDGAR | 0.63 | 0.92 | 17.87 | 2.72 |
| C | PKU | 0.64 | -3.96 | 16.91 | 2.65 |
| **C** | **REAS** | **0.61** | **-1.19** | **18.88** | **2.58** |
| A | PKU | 0.63 | -4.51 | 17.16 | 2.57 |
| E | PKU | 0.61 | -3.51 | 17.65 | 2.53 |
| D | PKU | 0.62 | -4.84 | 17.38 | 2.52 |
| C | FFDAS | 0.58 | -0.92 | 19.12 | 2.50 |
| C | CAMS | 0.59 | -2.77 | 18.06 | 2.49 |
| C | EDGAR | 0.58 | -1.71 | 18.53 | 2.47 |
| D | FFDAS | 0.58 | -1.69 | 19.19 | 2.44 |
| B | REAS | 0.58 | 1.44 | 20.14 | 2.41 |
| B | FFDAS | 0.60 | 2.97 | 20.19 | 2.41 |
| A | CAMS | 0.58 | -3.46 | 18.26 | 2.40 |
| A | EDGAR | 0.57 | -2.46 | 18.58 | 2.40 |
| A | FFDAS | 0.56 | -1.36 | 19.76 | 2.35 |
| C | MEIC | 0.63 | 5.15 | 20.68 | 2.34 |
| D | CAMS | 0.57 | -3.74 | 18.90 | 2.30 |
| D | EDGAR | 0.55 | -2.73 | 19.32 | 2.29 |
| E | REAS | 0.55 | -0.29 | 22.00 | 2.27 |
| A | REAS | 0.55 | -1.33 | 21.49 | 2.25 |
| A | MEIC | 0.59 | 4.60 | 21.84 | 2.20 |
| D | MEIC | 0.58 | 4.02 | 21.86 | 2.19 |
| D | REAS | 0.54 | -1.33 | 22.02 | 2.19 |
| E | FFDAS | 0.51 | 0.11 | 21.66 | 2.17 |
| E | CAMS | 0.52 | -2.21 | 20.93 | 2.13 |
| E | EDGAR | 0.51 | -1.11 | 21.72 | 2.09 |
| B | MEIC | 0.64 | 9.16 | 22.95 | 2.07 |
| E | MEIC | 0.57 | 5.94 | 23.34 | 1.99 |
| D | ODIAC | 0.52 | 3.63 | 22.57 | 1.96 |
| C | ODIAC | 0.53 | 5.04 | 22.80 | 1.91 |
| A | ODIAC | 0.49 | 4.56 | 24.46 | 1.69 |
| B | ODIAC | 0.54 | 8.63 | 24.91 | 1.63 |
| E | ODIAC | 0.47 | 5.81 | 25.67 | 1.51 |

**Table A1.** Statistical metrics for sensitivity tests, in situ $CO_2$ data at Xianghe. Unit of BIAS and RMSE is ppm. Rows shaded in dark gray indicate those where one or more statistical metrics surpass the thresholds defined in Eq. A1. Rows shaded in light gray represent combinations that are rejected due to the $XCO_2$ value falling outside the thresholds. The bold lines represent the final two options as determined by the methodology outlined in Appendix A.





| Test | Flux | CORR | BIAS | RMSE | S |
|------|------|------|------|------|------|
| **C** | **CAMS** | **0.52** | **2.19** | **206.50** | **2.81** |
| C | EDGAR | 0.52 | 19.09 | 208.24 | 2.67 |
| E | CAMS | 0.48 | 3.26 | 213.26 | 2.47 |
| A | CAMS | 0.45 | -7.09 | 210.84 | 2.31 |
| E | EDGAR | 0.48 | 22.33 | 216.09 | 2.28 |
| A | EDGAR | 0.46 | 12.50 | 213.59 | 2.27 |
| **B** | **CAMS** | **0.50** | **31.39** | **228.56** | **2.17** |
| B | EDGAR | 0.51 | 52.53 | 237.75 | 1.87 |
| D | EDGAR | 0.41 | 8.83 | 237.26 | 1.70 |
| D | CAMS | 0.39 | -9.19 | 237.31 | 1.60 |

**Table A2.** Same as Table A1 but for in situ $CH_4$. Unit of BIAS and RMSE is ppb.

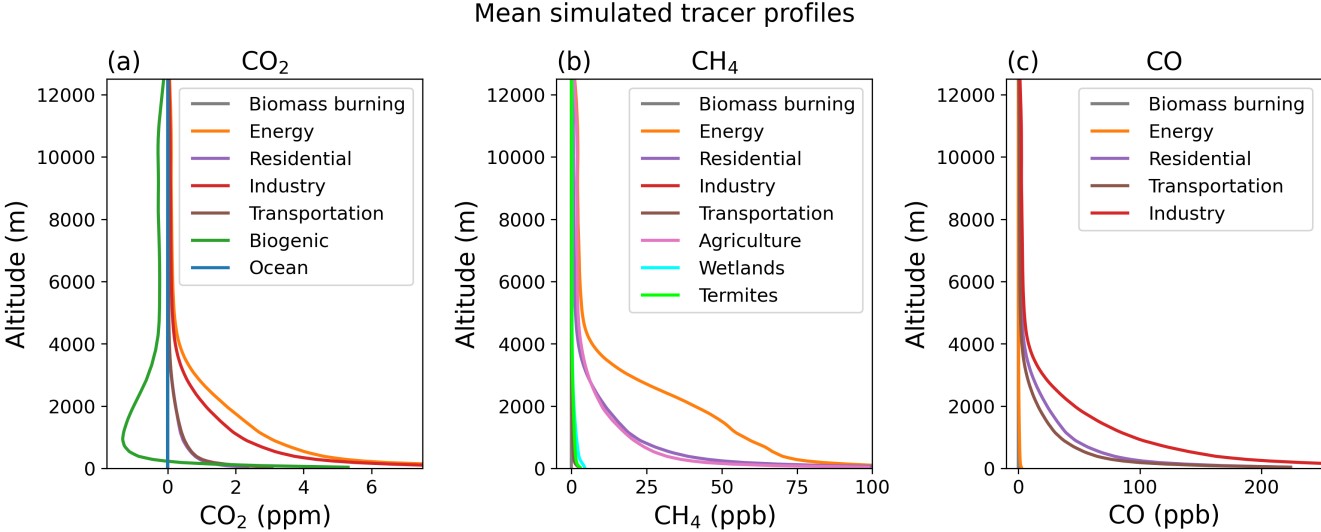

**Figure B1.** Mean vertical profile of the tracer fields in WRF-GHG for (a) $CO_2$, (b) $CH_4$ and (c) CO. All simulated hours were used for this plot.





| Test | Flux | CORR | BIAS | RMSE | S |
|------|------|------|------|------|------|
| D | MEIC | 0.77 | 0.62 | 1.52 | 2.87 |
| C | MEIC | 0.76 | 0.62 | 1.54 | 2.83 |
| E | MEIC | 0.66 | 0.95 | 1.95 | 1.86 |
| D | ODIAC | 0.78 | -1.27 | 2.28 | 1.86 |
| C | ODIAC | 0.79 | -1.32 | 2.29 | 1.85 |
| A | MEIC | 0.62 | 0.97 | 2.03 | 1.62 |
| D | FFDAS | 0.80 | -1.60 | 2.43 | 1.58 |
| C | FFDAS | 0.80 | -1.62 | 2.45 | 1.55 |
| E | ODIAC | 0.75 | -1.36 | 2.47 | 1.54 |
| D | EDGAR | 0.79 | -1.59 | 2.45 | 1.54 |
| A | ODIAC | 0.74 | -1.30 | 2.50 | 1.49 |
| B | ODIAC | 0.72 | -1.23 | 2.53 | 1.47 |
| C | EDGAR | 0.77 | -1.57 | 2.49 | 1.45 |
| D | CAMS | 0.79 | -1.70 | 2.52 | 1.38 |
| B | EDGAR | 0.76 | -1.54 | 2.55 | 1.38 |
| B | FFDAS | 0.75 | -1.52 | 2.58 | 1.33 |
| **C** | **REAS** | **0.80** | **-1.81** | **2.56** | **1.33** |
| E | FFDAS | 0.77 | -1.65 | 2.58 | 1.32 |
| D | REAS | 0.79 | -1.80 | 2.56 | 1.32 |
| E | EDGAR | 0.77 | -1.64 | 2.59 | 1.31 |
| C | CAMS | 0.77 | -1.68 | 2.57 | 1.30 |
| A | FFDAS | 0.76 | -1.62 | 2.60 | 1.28 |
| **B** | **CAMS** | **0.75** | **-1.64** | **2.63** | **1.22** |
| E | CAMS | 0.76 | -1.74 | 2.66 | 1.15 |
| D | PKU | 0.80 | -1.98 | 2.67 | 1.13 |
| C | PKU | 0.80 | -2.00 | 2.67 | 1.13 |
| B | REAS | 0.75 | -1.71 | 2.69 | 1.10 |
| A | EDGAR | 0.72 | -1.54 | 2.71 | 1.09 |
| E | REAS | 0.76 | -1.85 | 2.73 | 1.03 |
| A | CAMS | 0.72 | -1.65 | 2.76 | 0.98 |
| A | REAS | 0.75 | -1.84 | 2.75 | 0.97 |
| B | MEIC | 0.55 | 1.25 | 2.30 | 0.95 |
| B | PKU | 0.76 | -1.91 | 2.77 | 0.94 |
| E | PKU | 0.77 | -2.02 | 2.81 | 0.87 |
| A | PKU | 0.76 | -2.00 | 2.82 | 0.84 |

**Table A3.** Same as Table A1 but for $XCO_2$.





| Test | Flux | CORR | BIAS | RMSE | S |
|------|------|------|------|------|------|
| **B** | **CAMS** | **0.69** | **-0.79** | **20.53** | **2.94** |
| B | EDGAR | 0.69 | 0.65 | 20.94 | 2.62 |
| C | EDGAR | 0.67 | -0.96 | 21.24 | 1.73 |
| D | EDGAR | 0.66 | -0.80 | 21.45 | 1.47 |
| **C** | **CAMS** | **0.67** | **-2.16** | **21.31** | **1.12** |
| A | EDGAR | 0.65 | -1.17 | 21.72 | 0.86 |
| E | EDGAR | 0.65 | -1.66 | 21.76 | 0.59 |
| D | CAMS | 0.65 | -2.09 | 21.75 | 0.55 |
| A | CAMS | 0.65 | -2.75 | 21.45 | 0.37 |
| E | CAMS | 0.65 | -3.03 | 21.42 | 0.34 |

**Table A4.** Same as Table A1 but for $XCH_4$. Unit of BIAS and RMSE is ppb.

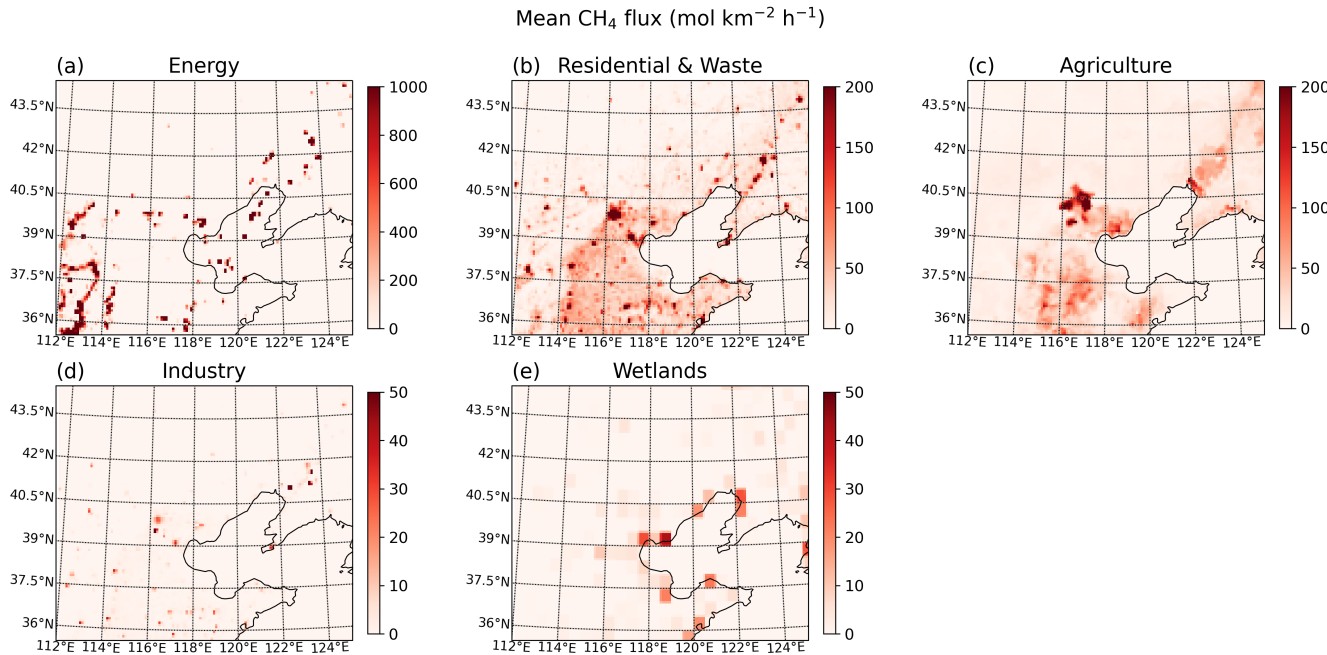

**Figure B2.** Map of the mean $CH_4$ flux (mol km$^{-2}$ h$^{-1}$) in WRF-GHG domain d02 during the entire simulation period for the most important sectors. Remark that different sectors have different ranges in the colorbar.



| Test | Flux | CORR | BIAS | RMSE | S |
|------|------|------|------|------|---|
| **B** | **REAS** | **0.78** | **-3.99** | **30.25** | **2.96** |
| B | PKU | 0.78 | -5.38 | 30.32 | 2.94 |
| D | REAS | 0.76 | -5.32 | 31.54 | 2.83 |
| E | REAS | 0.77 | -7.12 | 31.39 | 2.81 |
| **C** | **PKU** | **0.76** | **-6.88** | **31.73** | **2.79** |
| D | PKU | 0.76 | -6.82 | 31.75 | 2.79 |
| C | REAS | 0.75 | -5.53 | 32.00 | 2.79 |
| E | PKU | 0.77 | -7.85 | 31.74 | 2.77 |
| A | REAS | 0.75 | -7.15 | 32.35 | 2.72 |
| A | PKU | 0.75 | -7.51 | 32.67 | 2.71 |
| B | CAMS | 0.68 | -24.21 | 43.48 | 1.70 |
| E | CAMS | 0.66 | -25.16 | 44.35 | 1.61 |
| D | CAMS | 0.64 | -24.34 | 44.34 | 1.57 |
| E | EDGAR | 0.52 | 3.27 | 57.84 | 1.45 |
| A | CAMS | 0.59 | -23.19 | 44.78 | 1.44 |
| C | CAMS | 0.60 | -23.77 | 45.07 | 1.43 |
| B | EDGAR | 0.53 | 5.90 | 59.57 | 1.34 |
| A | EDGAR | 0.50 | 6.27 | 62.83 | 1.16 |
| D | MEIC | 0.65 | -37.13 | 49.72 | 1.05 |
| C | MEIC | 0.61 | -37.10 | 50.26 | 0.94 |
| B | MEIC | 0.53 | -30.94 | 47.80 | 0.93 |
| C | EDGAR | 0.46 | 8.30 | 67.22 | 0.87 |
| D | EDGAR | 0.47 | 8.66 | 68.01 | 0.86 |
| E | MEIC | 0.55 | -34.80 | 49.89 | 0.82 |
| A | MEIC | 0.52 | -34.49 | 50.35 | 0.72 |

**Table A5.** Same as Table A1 but for XCO. Unit of BIAS and RMSE is ppb.





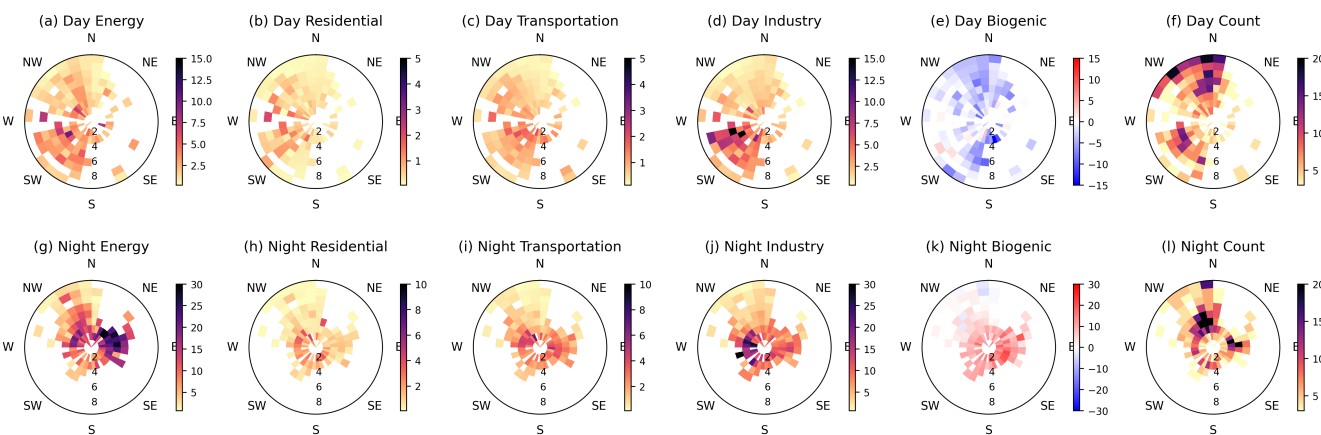

**Figure B3.** Same as Fig. 9 but for $CO_2$.

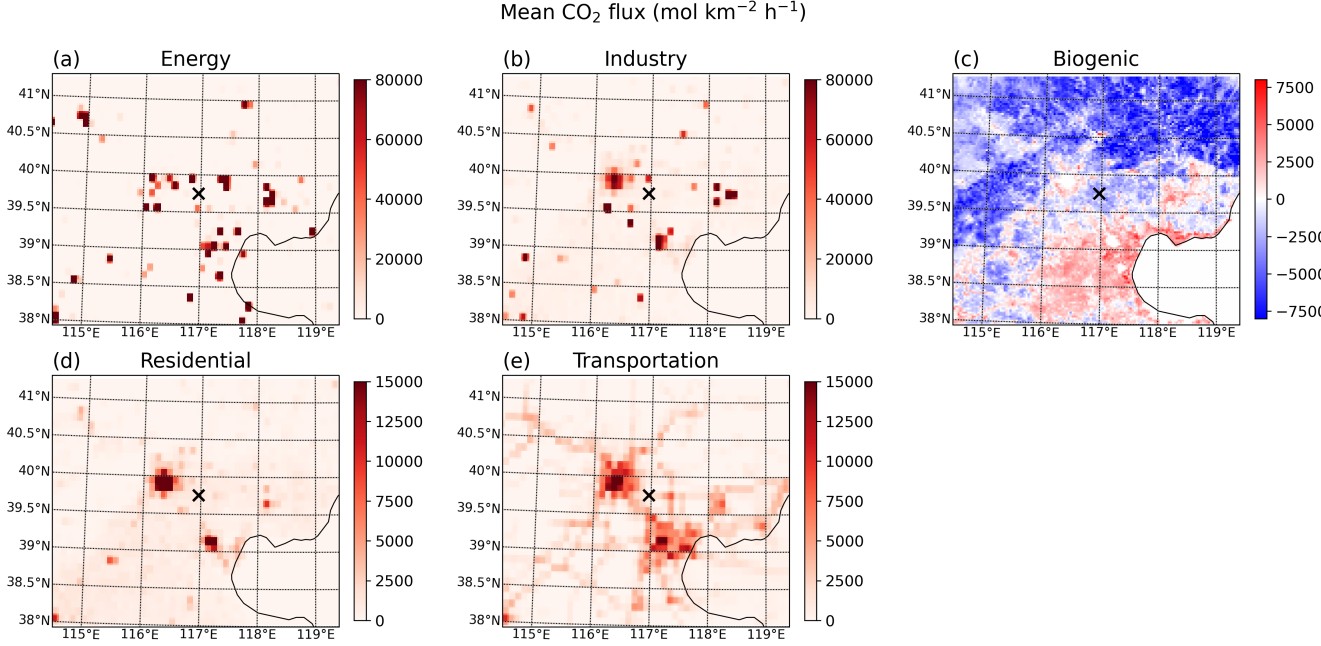

**Figure B4.** Same as Fig. 10 but for $CO_2$.



*Author contributions.* SC made the model simulations and performed the formal analysis, investigation and visualization. The research was conceptualized by SC, MDM and EM and supervised by MDM and EM. MZ, TW and PW have provided the observational in situ data at Xianghe. BL performed the L3 post-processing of TROPOMI and supported with computing tools to correctly compare the model with TCCON data. SC prepared the initial draft of this manuscript while it was reviewed and edited by MZ, BL, TW, MDM, EM and PW.


*Competing interests.* The contact author has declared that neither they nor their co-authors have any competing interests.

*Disclaimer.* The results contain modified Copernicus Climate Change Service information 2022. Neither the European Commission nor ECMWF is responsible for any use that may be made of the Copernicus information or data it contains.

*Acknowledgements.* We would like to thank all staff at the Xianghe site for operating the FTIR and PICARRO measurements. This work is supported by the National Natural Science Foundation of China (No. 42205140; 41975035). Emmanuel Mahieu is a senior research associate with the F.R.S.-FNRS. The authors acknowledge all providers of observational data and emission inventories. We thank the IT team at BIRA-IASB for their support on data storage and HPC maintenance. Christophe Gerbig, Roberto Kretschmer, and Thomas Koch (MPI BGC) are thanked for distributing the VPRM preprocessor code. Finally, we are grateful for fruitful discussions with Jean-François Müller (BIRA-IASB) and Bernard Heinesch (ULiège).




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
