# Peer review of "A WRF-Chem study on the variability of CO2, CH4 and CO concentrations at Xianghe, China supported by ground-based observations and TROPOMI"

_EGUsphere, 2023_

## Author Comment (AC1)

**Author comments on "A WRF-Chem study on the variability of $CO_2$, $CH_4$ and CO concentrations at Xianghe, China supported by ground-based observations and TROPOMI"**

We wish to thank the reviewers for their careful reading of our manuscript and their constructive comments. Their feedback has helped us to substantially improve our paper. We will start by addressing some issues that both reviewers identified, after which we carefully respond to each of the reviewers' remarks. The reviewer comments are written in black, while our author comments are in blue. Modifications to the manuscript are shown in *italic*. Note that new figures are included in this document and are referred to by upper-case letters, whereas others are referenced by their original number.

Furthermore, we suggest to slightly adapt the title to the following: "A WRF-Chem study on the TCCON column and in situ surface concentration of $CO_2$, $CH_4$ and CO at Xianghe, China".

**1 Concerns shared by both reviewers**

**1.1 Manuscript structure**

Both reviewers highlighted that the manuscript is difficult to follow in its current structure and should be reorganized. We admit that some improvements to the structure can be made. In fact, we propose to make the following modifications to the structure in the revised version.

First, we merged the section describing the Xianghe and TROPOMI observations with the model description section into one coherent 'Data, model and methods' section. The section describing how TROPOMI is compared to WRF-GHG (previously sect. 4) is also included in this section, as referred by the word 'methods' in the new section name. The comparison of WRF-GHG simulations with the Xianghe observations that was previously described in the part on the sensitivity tests has also been moved to a separate subsection here. The discussion about the sensitivity tests (previously section 3.2) was entirely moved to appendix A.

The earlier sections on model performance (Sect. 5), sector contributions (Sect. 6) and meteorological factors influencing variability (Sect. 8) have been moved to corresponding subsections of a new 'Results' section.

This section is followed by a new 'Discussion' section which includes four new subsections in which the observed model-data discrepancies are further analyzed. First, there is a subsection explaining the possible causes of the $XCO_2$ model underestimation during a large part of the simulation period. Similarly, there is also a separate subsection that attempts to clarify the seasonal $CH_4$ bias. Further, the old subsection 8.3.1 on the possible overestimation of the $CH_4$ emissions near Tangshan was also moved to this 'Discussion' section. A new subsection was created to assess the small nocturnal overestimation in the diurnal cycle of in situ $CO_2$. Finally, the subsection on meteorological factors that influence the Xianghe observations has been extended with a small case study explaining the driving processes behind a sudden increase in $XCO_2$ in July 2019. More detail on the new sections is given in the corresponding answers below. The final structure of the manuscript is now the following:

1. Introduction

2. Data, model and methods

   2.1. Xianghe site
   2.2. TROPOMI
   2.3. WRF-GHG modelling system

**1.2 $CO_2$ biases**

It was advised to further elaborate on the model-data discrepancies regarding $CO_2$ as the current coverage in the manuscript is rather limited. We have added two new subsections on $CO_2$ biases: one on the negative bias in the time series of $XCO_2$ from September 2018 until May 2019, and one on the slight model overestimation of the nighttime in situ $CO_2$ observed in the diurnal cycle.

The first subsection mainly consists of the lines 213-219 of the original version, complemented by more details about the CAMS-TCCON comparison:

*In Fig. 3a, we found that WRF-GHG is underestimating $XCO_2$ until May 2019 with about 2 ppm, after which the negative bias disappears. This bias is likely caused by the propagation of a similar error in the background data, a misrepresentation of the real sources and sinks in the region, or a combination of both. The CAMS validation report (Ramonet et al., 2021) presents "a very good agreement for all (TCCON) sites", suggesting that the CAMS reanalysis that is driving the WRF-GHG simulations is of good quality without known biases. However, their criteria for what constitutes "very good" appears to be relatively mild (within $\pm 2$ ppm). Moreover, the Xianghe site wasn't included in this report and the accompanying figure does not provide very detailed information. Therefore, we reproduce their analysis for several TCCON sites in the region, including Xianghe, Rikubetsu (43.46° N, 143.77° E), Tsukuba (36.05° N, 140.12° E), Saga (33.24° N, 130.29° E) and Hefei (31.91° N, 117.17° E) and for the period of our interest (September 2018 - September 2019), yielding Figure A. We find an underestimation of the CAMS reanalysis $XCO_2$ at all Asian TCCON sites from October 2018 until May 2019. More specifically for Xianghe, monthly mean errors range from*

[Figure]

Figure A: Monthly mean difference (in ppm) between CAMS reanalysis model and TCCON XCO$_2$ in Asia over the simulation period of this study.

*-2.20 (± 1.3) ppm in January 2019 to 3.38 (± 1.28) ppm in July 2019, which is of a similar order as the bias found with WRF-GHG (where the monthly mean differences with respect to the TCCON site of Xianghe range from -2.53 ± 1.7 ppm in December 2018 to 1.28 ± 1.57 ppm in July 2019). Therefore, we assume that the error pattern detected in the XCO$_2$ time series is primarily the result of the same pattern in the background information. Moreover, this bias pattern is not found in the in situ CO$_2$ time series (Fig. 3b), likely because the relative contribution from the background to the in situ concentrations is smaller than it is to the column data.*

The second subsection is new and discusses the possible causes for the nocturnal model overestimation for in situ CO$_2$:

*WRF-GHG shows an overestimation of surface CO$_2$ during a substantial portion of the night, while this is not the case for CH$_4$, implying a potential bias in the representation of CO$_2$ emissions rather than in the model dynamics. A possible source of error in the representation of CO$_2$ emissions might be in the parameterization of the VPRM parameters, which are taken from Li et al., 2020 but were optimized for the United States. An update on these parameters for China is essential to reduce the uncertainties on the biogenic CO$_2$ fluxes (Dong et al., 2021; Li et al., 2020). Furthermore, the simplicity of the linear respiration model could introduce inaccuracies, as suggested by Dong et al., 2021; X.-M. Hu et al., 2021; Li et al., 2020. Another potential error could arise from a misclassification of land cover type. VPRM fluxes are calculated as the weighted average of biogenic fluxes across seven vegetation classes, combined with fractional vegetation cover for every grid cell. This classification is based on the 1-km global land cover product SYNMAP by Jung et al., 2006, in which the region around the Xianghe site is represented by 100 % cropland. While this broadly aligns with the landscape, the omission of significant urbanization in this product, likely due to its dated nature, could lead to discrepancies. Given that built-up areas in WRF-GHG result in zero biogenic fluxes, this oversight could contribute to the observed overestimation of nighttime respiration and daytime photosynthesis near Xianghe. Finally, all emissions in WRF-GHG are released near the surface in the lowest model layer which is a simplification of reality since especially industrial and energy sources (power plants) usually emit CO$_2$ at higher altitudes. Brunner et al., 2019 showed that neglecting this fact could lead to an overestimation of the near surface CO$_2$ concentration.*

**1.3  Extending of WRF-GHG profiles above 50 hPa**

The WRF-GHG model simulates the atmosphere until 50 hPa, while the TCCON and TROPOMI instruments sense the complete atmosphere of the Earth. To facilitate the comparison of similar quantities, the model profiles were extended with a priori values when compared to the TCCON data. For our qualitative analysis using the TROPOMI data, instead of extending the profiles, we adjusted the color scale in the figures to account for this difference. Both reviewers questioned the latter approach, so we modified this analysis.

In our improved analysis, the model profiles are also extended with the TROPOMI a priori profiles before comparing with TROPOMI observations. Moreover, we smoothed these extended model profiles with the TROPOMI averaging kernels and regridded the resulting XCH$_4$ values to a common regular lat/lon grid. This improved approach allows a direct quantitative comparison between WRF-GHG and TROPOMI colocated data. More specifically, the description in the manuscript on how WRF-GHG is compared with TROPOMI is now the following:

*To compare the spatial XCH$_4$ distribution of TROPOMI with those of WRF-GHG, the model profiles are extended above 50 hPa with the TROPOMI a priori column number density profiles of CH$_4$ and dry air (mol $m^{-2}$) to ensure that both products in the comparison cover the same altitude range. Since a typical CH$_4$ profile shows a sharp decrease in the upper layers of the atmosphere, this part has a non-negligible impact on the column-averaged mole fraction. Further, the extended WRF-GHG CH$_4$ profiles are smoothed with the TROPOMI column averaging kernels and a priori profiles following Apituley et al., 2023. The column number density profiles of CH$_4$ and dry air are calculated from the hourly 3-D WRF-GHG output as follows:*

$$\rho_i^{CH_4} = \nu_i^{CH_4} \rho_i^{da}, \; with \; \rho_i^{da} = \frac{P_i}{RT_i} \frac{1}{1 + 1.6075 q_i} \tau_i. \tag{1}$$

*In the above equation $\nu_i^{CH_4}$ is the CH$_4$ dry air volume mixing ratio (ppb) and $\rho_i^{da}$ the dry air column number density in WRF-GHG layer i. The dry air column number density $\rho_i^{da}$ is calculated according to the ideal gas law, where $P_i$, $T_i$ and $q_i$ are the air pressure (Pa), temperature (K) and water vapour mixing ratio with respect to dry air (kg kg$^{-1}$), respectively. The thickness of layer i (m) is represented by $\tau_i$. Finally, R is the ideal gas constant 8.3145 J K$^{-1}$ mol$^{-1}$. Note that 1.6075 is the ratio of the molar mass of dry air with respect to the molar mass of water to convert wet air to dry air.*

*TROPOMI has an equator crossing time of around 13:30 local solar time, so we compute the equivalent simulated XCH$_4$ by taking the average over 12h-15h LT. Note that we use the model simulations from d02 (which has a horizontal resolution of 9×9 km$^2$) for this analysis, instead of d03 as for the comparisons with Xianghe observations, since a larger spatial extent is advantageous for a statistically effective comparison with TROPOMI.*

*Using the HARP toolset (part of the Atmospheric Toolbox, https://atmospherictoolbox.org/) for TROPOMI and the CDO software (Schulzweida, 2020) for WRF-GHG, both XCH$_4$ products are then binned to a common spatial grid to enable a quantitative analysis: we have chosen a regular latitude-longitude grid with a horizontal resolution of 0.05°.*

As a result of this new analysis, the corresponding figures (5 and 12 in the original manuscript) were updated and additionally a difference plot was included, see Figs. B and C. Overall, we find a mean bias of -11.96 ppb (TROPOMI - WRF-GHG, or -0.63% (TROPOMI - WRF-GHG)/WRF-GHG), in the studied region between 112°-120° E and 35.5°-40° N. This is in line with previous studies by Yang et al., 2020 and Tian et al., 2022 that concluded that TROPOMI XCH$_4$ is biased negatively with respect to TCCON at Xianghe. In fact, TROPOMI XCH$_4$ has been found to be slightly underestimating XCH$_4$ by 0.26% on average at most TCCON sites (Sha et al., 2021). Correspondingly, the paragraph in the section 'Seasonal CH$_4$ bias' was adjusted to as follows:

*The observed seasonal error pattern between the WRF-GHG CH$_4$ simulations and the observations at Xianghe could be due to one or more of the reasons mentioned previously. To gain a spatial perspective on this seasonal bias, we compared the WRF-GHG XCH$_4$ field with TROPOMI observations. Figure B presents the seasonal mean XCH$_4$ from both WRF-GHG and TROPOMI and their normalized difference over the broader Xianghe region. To emphasize seasonal variation, the mean bias over the entire simulation period (see Fig. Cd) was subtracted from the seasonal means, resulting in a "Normalized difference." Remark that we find a mean bias error of -11.96 ppb (or -0.63% (TROPOMI - WRF-GHG)/WRF-GHG)), which is in line with previous studies (Sha et al., 2021; Tian et al., 2022; Yang et al., 2020).*

*The general seasonal variation is similar to that observed in the comparisons with TCCON, with the largest model underestimation in summer and the largest overestimation in winter (see Fig. 3). This pattern is at least partially linked to similar biases in the boundary conditions, as discussed earlier. However, we cannot identify a distinct spatial pattern throughout the seasons that could point to errors within a specific source sector. Figure B shows differences on a large spatial scale, suggesting that the underestimation by WRF-GHG is linked to emission sources that are widespread in the region. Since the North China Plain is a livestock-*

*dominated region with strong urbanization and industrial activities, this implies that the fluxes of either agriculture (livestock), waste treatment, or both, rather than the fluxes from wetlands, are underestimated in summer in CAMS-GLOB-ANT. Given the lack of a clear outcome from our analysis, it is likely a combination of factors. This analysis additionally indicates an urgent need for more research on the seasonality of $CH_4$ emissions in north China.*

Similarly, the paragraph in 'Assessing $CH_4$ emission sources' was rewritten as follows:
*By comparing the yearly TROPOMI $XCH_4$ with WRF-GHG $XCH_4$, we want to assess if the $CH_4$ emissions from coal mines around Tangshan are indeed overestimated in CAMS-GLOB-ANT or not. Figure C shows the maps of the mean $XCH_4$ during the entire simulation period: September 2018 until September 2019. The yearly mean total $CH_4$ fluxes from CAMS-GLOB-ANT in the WRF-GHG domain 2 is also given, as well as the difference between WRF-GHG and TROPOMI. . . . We observe slightly elevated $XCH_4$ values near the coal mines of Tangshan in both the WRF-GHG (1887.84 ppb compared to 1884.49 ppb in the surrounding area) and TROPOMI (1879.01 ppb vs 1876.64 ppb) maps. The surrounding area is defined between 39.3-40 °N and 117.8-118.8 °E as there are no major $CH_4$ sources located therein, while the coal mine sources are concentrated in the area between 39.45-39.8 °N and 118.15-118.6 °E. Although these differences are minor, the enhancement in WRF-GHG is somewhat greater than in TROPOMI. More specific, the mean difference between WRF-GHG and TROPOMI is 7.87 ppb around Tangshan, while it is 8.81 ppb near the emission sources, suggesting that the model overestimation is more pronounced over the coal mines of Tangshan compared to the surrounding area. This difference is statistically significant with a p-value of 0.004, according to a one-sample t-test. This analysis suggests that these emission sources are indeed overestimated in CAMS-GLOB-ANT, occasionally leading to an overestimation of the energy tracer at Xianghe.*
*Note that the $XCH_4$ maps in Fig. C suggest that it is very likely that the $CH_4$ hotspot around 36.25°N, 116.75°E is overestimated as well, as the very strong $XCH_4$ enhancement in WRF-GHG is absent in the TROPOMI map.*

**2 reviewer 1**

**Major Concerns**:

The literature review is narrowly focused on recent studies, with minimal references before 2017. This approach overlooks significant earlier research, especially in GHG model-data comparisons in urban settings, as seen in projects like INFLUX, LA Megacity, and European initiatives like Mega-Paris and COBRA.
We are familiar with the references mentioned by the reviewer, however, we feel like these urban studies are less relevant for the current research. Especially, since our objectives are not aimed at constraining an urban $CO_2$ budget, but rather to exploit the measurements at a specific site (that is indeed situated in an urbanized area) and reveal the source sectors and meteorological factors that contribute to the observed variations. Therefore, our literature review focuses on the observations at Xianghe, which form the center of our study. Subsequently, we explore studies utilizing the WRF-Chem model to investigate greenhouse gases in China and beyond, emphasizing its wide-ranging applications and robust performance in analyzing similar observations: ground-based in situ and FTIR data of $CO_2$, $CH_4$ and CO. The currently referenced studies are thus essential in supporting the chosen methodological approach for analyzing the observations. The modeling efforts of Lac et al., 2013 in the Mega-Paris project, on the other hand, utilize the Meso-NH model rather than WRF-Chem. Further, we did not encounter studies combining data from the COBRA project with the WRF model (assuming the reviewer refers to the $CO_2$ Budget and Rectification Airborne study over North America in 2000 (Gerbig et al., 2003)). Therefore, we have solely included references from the INFLUX and LA Megacity projects, which apply WRF to exploit high resolution $CO_2$ observations from urban campaigns in the United States:
*Elsewhere, the WRF model has demonstrated efficacy in analyzing comparable observations (Callewaert et al., 2022; X.-M. Hu et al., 2020; Park et al., 2020; Zhao et al., 2019), as well as high-resolution $CO_2$ measurements in urban settings (Feng et al., 2016; Lauvaux et al., 2016).*

The study's sensitivity tests concentrate on PBL, surface layer, and radiation schemes, but the rationale for this focus is unclear. Notably, the land surface model variety, crucial for PBL variations, is absent, which is

[Figure]

Figure B: Seasonal mean XCH$_4$ (ppb) over the domain d02 (provinces of Beijing, Tianjin, Hebei, Shanxi and part of Shandong) as simulated by WRF-GHG (first column) and observed by TROPOMI (second column), as well as the normalized difference between them (WRF-GHG - TROPOMI, in ppb). Normalized difference indicates that the mean difference over the entire simulation period is subtracted from the seasonal means. The seasons are defined as (a,b,c) SON: September - November (autumn), (d,e,f) DJF: December - February (winter), (g,h,i) MAM: March - April (spring) and (j,k,l) JJA: June - August (summer). White pixels indicate that there are no observations available during the entire period.

[Figure]

Figure C: (a) The $CH_4$ flux from all sectors in CAMS-GLOB-ANT averaged from September 2018 until September 2019 and regridded to WRF-GHG grid d02 (9 km resolution). Mean $XCH_4$ over the same period as (b) simulated by WRF-GHG and (c) observed by TROPOMI (both regridded to 0.05 °). (d) Mean difference between WRF-GHG and TROPOMI $XCH_4$ over the entire simulation period.

a considerable oversight in the experimental design. See Díaz-Isaac et al., 2018.

The authors are aware of the wide variety of physics parameterization schemes that are available in the WRF modeling system. The sensitivity tests were performed in an attempt to find a set of model options that performs reasonably well without carrying out a full analysis of all existing options (which was both not the scope of the study and not feasible in the context of computing time and timeframe of the study). To decide which parameterization schemes were tested, we looked at studies using WRF to simulate greenhouse gases in China, as performance of these schemes are highly location and application dependent. While there was a wide range in PBL, surface layer and radiation schemes, almost all examined studies used the Noah land surface scheme (Che et al., 2022; Dong et al., 2021; C. Hu et al., 2018; Li et al., 2020; Liu et al., 2018). We therefore adopted this latter parameter scheme, supposing that it is the most appropriate one in our case (even though changing it might have a non-negligible impact on the results as your reference showed). Moreover, it is important to note that in flux-inversion studies based on in situ observations, such as those referenced by Díaz-Isaac et al., 2018, high model accuracy for in situ observations is essential, while this level of accuracy is not required in our current study, which aims to identify the main sectors and meteorological factors contributing to the measured variability. Finally, it should be noted that the discussion of the sensitivity tests was moved to the appendix, following feedback from the other reviewer. The introduction of this appendix is now the following:

*Sensitivity tests were carried out to identify a model configuration that matches the observations well. We have tested several physical parameterization schemes and anthropogenic fluxes because these elements are essential to accurately simulate tracer concentrations. The initial set of physical parameterization schemes (BASE) was taken from Li et al., 2020 and Dong et al., 2021 as they have shown good model performance for simulating $CO_2$ concentrations in China. Four alternative combinations (A-D) were created by changing the schemes for the longwave and shortwave radiation, planetary boundary layer (PBL) and surface layer physics, leading to 5 different model configurations in total (see Table A1). Remark that there are several more physical parameterization schemes that could have been included in these tests. Nevertheless, a full sensitivity analysis is outside the scope of this study. Thus, we restricted our tests to the most frequently used*

*schemes in the literature and chose the combination that produced satisfactory model simulations without additional optimization.*

The method to compensate for WRF-Chem's limitation at 50 hPa for $XCO_2$ calculations involves integrating TCCON a priori profiles. A more consistent approach would be to use the global-modeled $CO_2$ that provides lateral boundary conditions, here CAMS values, above 50 hPa, ensuring a fairer comparison with TROPOMI results. See Butler et al., 2020.

First, we want to highlight that the comparison with TROPOMI $XCH_4$ in the revised manuscript has been adjusted, as elaborated above in Sect. 1.3, and now includes extended model profiles (using the TROPOMI a priori profiles). In this regard, we believe our approach is consistent. We appreciate your suggestion to use globally modeled concentrations for lateral boundary conditions above 50 hPa as an alternative method. However, we consider this approach to be equally viable but not necessarily more consistent.

It is also important to clarify that the TROPOMI comparison focuses on $XCH_4$, not $XCO_2$. For $XCO_2$, concentrations above 50 hPa have a negligible impact on the column-averaged mole fraction, making the type of data used to extend the profiles less critical. In contrast, for $XCH_4$ information above 50 hPa is relevant due to the typical sharp decrease in concentration around the stratosphere. Therefore, using CAMS data above 50 hPa for $CH_4$ could be a reasonable and consistent approach, when applied to both TCCON and TROPOMI comparisons. However, the accuracy issues with CAMS $CH_4$ data in the stratosphere are well-documented (Agustí-Panareda et al., 2023; Ramonet et al., 2021), and would introduce known biases into our study. Furthermore, given that our study is primarily focusing on one specific site (Xianghe), extending the profiles with the TCCON a priori data for that site is a logical choice. This a priori data is optimized to be more accurate than the global model data from CAMS, particularly for localized analyses (Laughner et al., 2023). We have added the following sentence to the (new) subsection describing how WRF-GHG is compared with the TCCON data:

*Note that the optimized a priori profiles of the TCCON GGG2020 data show improved accuracy in the lower stratosphere (Laughner et al., 2023), supporting our decision to utilize this data for extending the model profiles.*

The model's performance is somewhat overstated. For example, a mean bias of -1.43 ppm in $XCO_2$ is significant. The high $CO_2$ correlation coefficients are likely influenced by seasonal variation, necessitating detrending for accurate assessment. Additionally, the model's capture of the $CO_2$ diurnal cycle shows notable discrepancies, especially during nighttime. This also reflects the lack of a literature review on the authors' end.

The 'large' bias of -1.43 ppm for $XCO_2$ is mainly caused by the mentioned underestimation of 2 ppm during a large part of the simulation period, which can primarily be attributed to the lateral boundary conditions (lines 213-219). By restructuring the manuscript, as described in the beginning of this document, we created a separate subsection to discuss this bias more in depth (see Sect 1.1 and 1.2).

Regarding the high $CO_2$ correlation coefficients, indeed this is likely influenced by seasonal variation. We recalculated the correlation coefficient on the residuals of the time series with respect to their sinusoidal fit following :

$$\mathbf{Y(t)} = A_0 + A_1 \cdot \mathbf{t} + \sum_{k=1}^{3} \left( A_{2k} \cos(2k\pi\mathbf{t}) + A_{2k+1} \sin(2k\pi\mathbf{t}) \right) + \epsilon(\mathbf{t}).$$

The value becomes 0.70 instead of the initially reported 0.85 and is adjusted in the manuscript as such. Moreover we have added the following sentence to the caption of the statistics table (previously Table 3): *Note that the $XCO_2$ time series was detrended before calculating the correlation coefficient in order to remove the effect of the seasonal variation.*

Finally, we have also expanded our discussion on the $CO_2$ diurnal cycle regarding the nocturnal overestimation. This can be found in a new subsection, as detailed above (Sect. 1.2). By adding the modifications described above, three references were added to the bibliography of the manuscript.

The authors' conclusions about error sources, particularly the underestimation of $XCO_2$, seem speculative and

lack a detailed description of the model configuration, including aspects like data assimilation and restarts. Without the info, it's difficult for me to judge if the authors' interpretation makes sense or not.

As detailed above, the underestimation of $XCO_2$ is now elaborated in a new subsection with more detail. Further, we added more information about the model configuration in the model description section:

*The simulations are re-initialized with the ECMWF ERA5 data every 30 h, starting at 18:00 UTC the previous day with a 6 h spin-up period, as done in other WRF-GHG modelling studies (Feng et al., 2016; Park et al., 2018; Pillai et al., 2011). Every day at 00:00 UTC, the tracer fields from the previous run are copied to the new simulation to ensure continuous transport of the concentrations.*

No other data assimilation or nudging procedures are applied.

A Lagrangian approach might be more suitable for tracing back signals to their sources and sinks, as the current Eulerian perspective may not effectively disentangle these signals.

Indeed, a Lagrangian approach might be more suitable for the specific purpose of tracing back parcels to their sources and sinks. However, one of the objectives of this study was to assess the usefulness of the WRF-GHG model to analyze the observations, which goes beyond tracing back to sources and sinks only. Moreover the current Eulerian approach provides a direct comparison with both the surface and column observations simultaneously, which is less straightforward for Lagrangian models. Additionally, the possibility to separate the simulated signal into different tracers and link it with the simultaneously modeled meteorology are some important benefits of WRF-GHG with respect to Lagrangian models. An additional study that would focus on the specific geographical areas that influence the signal at Xianghe, can be carried out with WRF-GHG by defining appropriate model tracers, however for that specific purpose, a Lagrangian approach might indeed be more suitable.

The manuscript suffers from poor organization and numerous language and grammatical errors, making it challenging to follow. For instance, the placement of CO discussion in the $CO_2$ section and the absence of a dedicated section for $CO_2$ biases, despite their significance, are confusing.

Following the comments of both reviewers, we thoroughly restructured the manuscript (see Sect. 1.1), including a dedicated section for $CO_2$ biases (see Sect. 1.2). Regarding the language and grammatical errors, we are happy to correct them if we get more precise indications.

**Specific Comments:**

Line 67: Expand citations to include key studies like Feng et al., 2016 (ACP), Lauvaux et al., 2016 (JGR-A), etc. See my comments above regarding those urban projects for more references.

As stated above, those references have been added to the introduction.

Line 105: Reevaluate the relevance of the Fast et al. (2006) citation.

It is requested by the WRF-Chem community to add this citation in any publication. See the WRF-Chem User's Guide (p 43): "When presenting, or publishing results from studies using the WRF-Chem model, it is requested that you cite the Grell et al. (2005) and Fast et al. (2006) manuscripts provided in the relevant publications section of this chapter. For any application that uses the indirect effect, please also cite Gustafson et al. (2007). And likewise, when using other significant features in the WRF-Chem model, the user should examine the reference list on the WRF-Chem web page and cite the developer's paper(s) (https://ruc.noaa.gov/wrf/wrf-chem/References/WRF-Chem.references.htm)."

Line 165-167: Clarify the intended message for better understanding.

The goal is to find a model configuration that is appropriate to simulate all the different observations at Xianghe. Both the choice of physical parameterization schemes and anthropogenic flux inventories have a large impact on the simulated mole fractions. Therefore, we aim to select that combination of physical parameterization schemes and flux inventories that matches well with the observations, both near the surface and in the total column and for the three considered species. If a specific model configuration is best suited to -for example- simulate the surface $CO_2$ mole fractions, it is not necessarily the most optimal choice for the $XCH_4$ observations. Moreover, we have to take into account that the choice of physical parameterization schemes must be the same for all three species, but the anthropogenic flux inventories might differ from

species to species. On line 165-167 we try to explain that there is not a single combination that results in model simulations that are closest to the measurements (highest score S), for all observation types. We propose to change lines 165-167 (which are moved to the appendix in the new structure, see Sect. 1.1) as follows:

*Unfortunately, there is not one combination of physical parameterization schemes and anthropogenic flux inventories that yields optimal scores for all species ($CO_2$, $CH_4$ and CO) across various observation types (surface and column). To identify the most appropriate model configuration for simulating all observations at the Xianghe site, it is necessary that the chosen physical parameterization schemes (denoted as test A - D, BASE) show satisfactory skill scores across all five time series. Moreover, the choice of anthropogenic flux inventory, although potentially varying among species, should yield reasonable score values for all observation types of the same species.*

Line 186: Correct the statement about TROPOMI's orbit.
We adapted line 186 to:
*TROPOMI has an equator crossing time of around 13:30 local solar time.*

Line 201: Provide justification for the chosen two-week spin-up period. My experience is that sometime after 20 days, I still see influences from initial conditions.
The spin-up period is mainly chosen to avoid numerical artefacts arising from the initialization process and to ensure some initial mixing of the surface-atmosphere fluxes within the different tracer fields. For the first (avoiding numerical artefacts), a period of a couple of hours is typically sufficient. For the second (ensuring initial mixing), this period is loosely based on the amount of time a typical weather system needs to travel from one model boundary to the next. After that period, the initial boundary conditions and surface fluxes should be well-mixed throughout the model domain. In fact, a couple of days is very likely already sufficient to achieve this purpose. The period of two weeks is therefore chosen rather conservatively to be on the safe side. We modified the following sentences on lines 200-202:
*However, to ensure a sufficient mixing of the surface-atmosphere fluxes and boundary conditions over the entire model domain, the first two weeks were regarded as a spin-up phase. Therefore, the analysis is made on one full year of data: from 1 September 2018 until 1 September 2019.*

Figure 6: Narrow down the wind direction ranges for NW and SW.
As the comment and its motivation are not entirely clear to the authors, we assume the reviewer wants to see the analysis on a subset of the data with the 'pure' southwest and northwest winds, i.e., excluding the data points where the wind is coming exactly from the south, west or north. Therefore, we redefined the NW category as winds with an angle between 292.5° and 337.5°, and SW winds with an angle from 202.5° to 247.5°, yielding the updated Figure 6 (see Fig. D below). The main conclusions remain unaltered: higher concentrations coincide with 800 hPa winds coming from the SW while NW winds correspond with lower concentrations. Furthermore, the low p-values indicate that the differences remain statistically significant.

Figure 7: high correlations after diurnal cycle removal are expected in terms of synoptic scale variations.
The fact that the high correlations are related to synoptic scale variations is exactly the point that is being made here. However we agree that the added value of this figure is rather small, so we removed it in the revised manuscript.

Line 333: there is a surface layer under the PBL.
Unfortunately, this comment is not understood by the authors and could therefore not be addressed. There is nothing mentioned about the PBL on line 333, nor in the corresponding section.

**3   reviewer 2**

**Major Concerns**:

[Figure]

Figure D: Updated Fig. 6 from the original manuscript, where the wind direction categories are as follows: NW, winds with an angle between $292.5°$ and $337.5°$ and SW, winds with an angle between $202.5°$ to $247.5°$. There are 72 days with NW winds and 33 days with SW winds.

The unorthodox structure of the text makes it at times difficult to follow. The flow of information seems to reflect way the experiments were performed, describing the 'results > analysis > extra-results-to-test-hypothesis-X > analysis' chain, but this might not be the best choice for the final publication. I strongly recommend that the authors streamline and clarify the text structure.
The structure of the manuscript was altered. More detail is given in Sect. 1.1 above.

In multiple places in the manuscript, the quantitative analysis and/or information is needed and is not given. For example, in L213 the authors write that "a similar pattern (of negative bias) was found when comparing CAMS reanalysis data set with the TCCON data at Xianghe and other sites in that part of the globe", but do not provide neither the size of that bias nor the reference to the study where it was published. Another example, in L404: "We find slightly elevated $XCH_4$ values nearby the coal mines of Tangshan ($6°N$, $118.4°E$) in both WRF-GHG and TROPOMI maps" – it is unclear what does "slightly elevated" mean here? Numeric value here would be objective and reproducible. I've listed some of the places where more quantitative approach is needed below (see "Detailed comments").
As explained in the 'Shared concerns' section above (Sect. 1.1 and 1.2), we have added more detail on the $XCO_2$ bias in a new subsection, as well as on the $CH_4$ bias. Further, numeric values have been added to the section on sector contributions for all species, including a new table. Additionally, the discussion on the diurnal cycle of near surface concentrations was enhanced. More detail on these specific modifications to the manuscript are given in the reactions to the respective comments. Finally, the methodology to compare WRF-GHG with TROPOMI $XCH_4$ was adjusted to make a quantitative analysis possible, as was explained in Sect. 1.3.

Seasonal bias of $CH_4$ is heavily discussed, but the authors do not attempt to quantify how much of this effect is related to background variability, and how much is local. While this doesn't allow for direct data to model comparison, this could still inform on relevance of regional vs local influences.
In the newly structured manuscript, there is a separate section discussing the $CH_4$ bias (see Sect. 1.1). Similarly as for the analysis of the $XCO_2$ bias, we calculated the CAMS bias with respect to other TCCON sites in the region and included the results in this section. This shows that both the overestimation in winter and the underestimation in summer is larger for WRF-GHG than it is for CAMS. We therefore conclude that the seasonal bias of $CH_4$ is caused by both local (incorrect emissions) and regional (biases in background information) influences. We have added Fig. E and the following description in the manuscript:
*The CAMS validation report by Ramonet et al., 2021 found a seasonality in the bias between CAMS $CH_4$ and TCCON. To have a closer look at the pattern in our simulation and include Xianghe in the analysis, we reproduce their calculations for several TCCON sites in the East-Asian region (Rikubetsu, Xianghe, Tsukuba, Saga and Hefei) and for the period of our interest, as shown in Fig. E. Indeed, we find a seasonal bias*

*where CAMS is overestimating TCCON from December until May and showing a small underestimation in the rest of the period. The bias at Xianghe ranges from 13.17 ppb in February 2019 to -6.56 ppb in August 2019 (monthly mean differences). The monthly mean bias of WRF on the other hand, ranges between 26.32 ppb in February 2019 and -28.82 ppb in August 2019 and shows a significantly larger amplitude than the CAMS bias. Moreover, the same seasonal pattern is found in the time series of the differences for the in situ data (Fig. 3d). Hence, the seasonal bias between WRF-GHG simulations and local observations of $XCH_4$ at Xianghe likely arises from biases in the background information as well as errors in the seasonality of the $CH_4$ emissions.*

[Figure]

Figure E: Monthly mean difference (in ppb) between CAMS reanalysis model and TCCON $XCH_4$ in Asia over the simulation period of this study.

Discussion on biases for $CO_2$ is limited to 7 (seven) lines and focuses on the jump of summer of 2019. Looking at the data quality makes me wonder about stability of the measurement system at that time (especially considering concurrent gap in the insitu system - was that related?), but regardless of that, the latter part of the data is ignored, the $CO_2$ discrepancy is still worth investigating.

*As mentioned above (Sect. 1.2), we have included a separate section discussing the $CO_2$ bias in the revised manuscript. However, this mainly focuses on the model underestimation over the first part of the simulation period (September 2018 - May 2019) and not on the jump of Summer 2019.*

*During these months, the WRF-GHG performs actually quite well, so to describe this part of the data as a 'discrepancy' is rather confusing. We assume the reviewer is wondering about the sudden increase in $XCO_2$ in July 2019 compared to the weeks before and after (which we will refer to by 'summer spike'). First of all, we want to clarify that the in situ $CO_2$ and $XCO_2$ time series are observed by two completely independent measurement systems, so a defect in one does not affect the stability of the other one, unless ofcourse it is related to atmospheric conditions. However, the gap in the in situ $CO_2$ time series between 29-07-2019 and 21-08-2019 is due to malfunctions of the instrument (Yang et al., 2021). Nevertheless, this $XCO_2$ summer spike is indeed worth investigating and deserves to be mentioned in the manuscript. We therefore added a subsection to explain this specific event:*

*Between 20-29 July, a notable spike in $XCO_2$ levels is observed (see Fig. 2a), diverging from the typical decreasing trend of $XCO_2$ from May to September, which is linked to the northern hemisphere's growing season and increased photosynthesis. We will focus on the model simulations between 7 July 2019 and 30 August 2019 to explain the causes of this $XCO_2$ summer spike, as WRF-GHG correlates well with the observations during this period (correlation coefficient of 0.84). The total simulated $XCO_2$ increases from 407.43 ($\pm$ 1.19) ppm before to 410.76 ($\pm$ 1.19) ppm during the summer spike (20-29 July), and then decreases to 405.75 ($\pm$ 0.98) ppm after, as shown in Fig. Fa. This sudden increase of more than 3 ppm is significant and warrants further investigation.*

*Figure F shows the simulated background and tracer contributions during this period. Figure Fa shows that the background $XCO_2$ remains relatively constant in July (408.69 $\pm$ 0.78 ppm), and decreases to 406.67 $\pm$*

[Figure]

Figure F: Simulated time series of XCO$_2$ at Xianghe from 7 July 2019 to 30 August 2019, with the spike period highlighted in all panels. (a) Background tracer (cyan) and total tracers (black) from WRF-GHG at Xianghe, with hourly values shown as thin lines and points for TCCON observation times. Daily mean 800 hPa wind direction is indicated by wind barbs at the bottom. (b) Time series of different tracer contributions at Xianghe, with hourly values as thin lines and observation times highlighted. (c) Colour coded vertical profiles of the biogenic CO$_2$ contributions (left y-axis) shown in red and blue, and surface temperature (right y-axis) in black.

*0.73 ppm in August. It clearly indicates an enhancement of the tracers from below the background before and after the summer spike to above the background during the spike period. Looking at the different tracers in Fig. Fb, we see that it is mainly the biogenic tracer that has a different behaviour in the spike period compared to the periods before and after. Thus, the increase in $XCO_2$ between 20-29 July is mainly linked to a less strong biogenic sink (-0.33 ± 0.57 ppm) compared to the periods before (-2.88 ± 0.71 ppm) and after (-1.76 ± 1.11 ppm). Further analysis reveals that during the spike, a heatwave with surface temperatures up to 39° C occurred, together with 800hPa winds predominantly from the west (see Fig. Fa and c). The biogenic tracer also shows increased values across a large vertical extent in the troposphere (Fig. Fc), indicating advection from other regions. Synoptic maps (not shown) suggest eastward advection of a warm air mass with high biogenic $CO_2$ levels, originating from the Gobi Desert and grasslands in Inner Mongolia, both areas characterized by sparse vegetation and elevated temperatures. Additionally, the mean biogenic $CO_2$ flux around Xianghe, as calculated by VPRM, is slightly higher between 20 and 29 July compared to the periods before and after. In VPRM, the respiration component is linearly dependent on surface temperature, and the gross ecosystem exchange also has a temperature dependency representing the temperature sensitivity of photosynthesis, with $CO_2$ uptake decreasing at temperatures higher than optimal (Mahadevan et al., 2008). Indeed, it has been shown that extreme temperatures impact $CO_2$ fluxes (Gupta et al., 2021; Ramonet et al., 2020; Xu et al., 2020). Therefore, we conclude that the spike was caused by an atmospheric circulation anomaly resulting in the advection of a warm air mass with high biogenic $CO_2$ levels, along with a locally reduced ecosystem exchange due to the resulting hot temperatures.*

**Specific Comments:**

L4: passive tracer is also an option from base WRF. In order to avoid confusing GHG module with that separate module, I suggest: "are produced by model's greenhouse gas module WRF-GHG"
We have adopted this suggestion in both the abstract and the model description section.

L5: add correlation coeff. also for $CH_4$ for completeness
Ok, added.

L9: 'sectors' are usually used for anthropogenic activities, 'biosphere sector' is rarely used (if at all). Consider using 'biosphere fluxes' or 'biospheric activity' / 'NEE (Net Ecosystem Exchange)' instead throughout the text.
We adjusted the sentence in the abstract to the following:
*For $CO_2$ the industry and energy sectors, together with the biosphere are found to be the primary contributors to the total simulated concentration*

L10: 'Residential & waste' is meant as a clumped sector as defined in this work, but it requires reading the paper to understand its usage in the abstract. Rephrase / clarify.
In the abstract, we have adjusted the sentence to:
*... whereas $CH_4$ concentrations are predominantly attributed to activities from the energy, agriculture, residential heating and waste management sectors.*
Then in the first occurrence of 'residential & waste' in the main body we have added a small clarification:
*(which combines both residential heating and waste management sectors).*

L32: "also been rising for the last 200 years"
Ok, adjusted.

L63: It's sufficient to ignore chemical reactions change if the residence time in the limited area domain is short enough that the change in mole fractions is smaller than the precision of the measurements. It can be shown for $CH_4$, however, assuming exponential decay, over five days it can already decay by over 2 ppb, and depending on the domain size the residence time of air can be even longer, depends on the circulation. In most of the cases this can be ignored, but I believe this should be addressed in here for both $CH_4$ and CO, because the domain is relatively large. It should also be discussed later in the paper, where bias of $CH_4$ is described, as the effect of omitting the CH4 destruction could have some effect on observed differences.

In the introduction we have added the following:
*..., which is generally a valid assumption regarding the regional domain and the relatively long atmospheric lifetimes of the target species ($\sim$ 100 yrs for $CO_2$, $\sim$ 10 yrs for $CH_4$ and several weeks for CO)(Dekker et al., 2017). Nevertheless, both $CH_4$ and CO are prone to chemical reactions in the atmosphere, making this assumption a simplification of actual conditions, which should be taken into account when analyzing the results.*
Furthermore, we have clarified this fact in the model description section:
*WRF-GHG is a Eulerian atmospheric transport model that simulates the 3-D concentration of trace gases at every time step simultaneously with meteorological fields, neglecting chemical reactions.*
Finally, we have included the following sentence in the conclusions:
*Additionally, integrating exponential decay for $CH_4$ in future model adaptations to represent the OH sink could enhance model accuracy while avoiding more computationally demanding chemistry simulations.*

L78: Please provide more exact location of the site, at the moment this points to a random location when checked on publicly available mapping portals.
We have changed the precision to 39.7536 N, 116.96155 E.

L88: I've never heard about calling CO 'Xgas' before. While interesting, I would suggest removing this.
Xgas is the general abbreviation of the total column-averaged dry air mole fraction, not just of CO, but for any species. It is then indicated by an 'X' in front: $XCH_4$, $XCO_2$ and XCO. To avoid confusion, we changed this sentence to:
*... providing total column-averaged dry air mole fractions (the so-called Xgas) of $CO_2$, $CH_4$ and CO.*

L91: The instrument is installed on a tower, or is there tubing coming from the tower? Please clarify.
As described in Yang et al., 2021 (Section 2.2), the sampled air is introduced into a tube. To avoid confusion we altered the instrument description in the manuscript to the following:
*The instrument samples air from an inlet fixed at 60 m above the ground on a tower.*

L95: typing error in Sentinel-5 ('-nal')
Ok, adjusted.

L97-99: "In our study, we use...". Please add appropriate reference to the dataset. Same place: Also, evaluation of L2 product is discussed but what about evaluation of L3 that is used here?
In our original analysis, we used the L3 data that is internally available at BIRA-IASB for validation purposes. This dataset can also be viewed at https://terrascope.be, but does not have a reference. There is however an algorithm theoretical base document describing this data, also available at the Terrascope platform or here. It is described that the L3 product was generated using the HARP software, which is part of the ESA Atmospheric Toolbox project. However, this particular L3 product does not contain the a priori profiles or column averaging kernels (AK) that we wanted to use in our new approach (see Sect. 1.3 above). Therefore, we now use the official L2 product ("TROPOMI Level 2 Methane Total Column Products. Version 02", 2021) directly, and create a new L3 product with HARP that includes the prior and AK information, while still following the same approach as explained in the ATBD. As such, only the L2 product is referenced in the revised manuscript, and the mention of the L3 dataset is omitted.

L101: -0.6 % and -0.39 % translate into roughly 11 and 7 ppb, respectively. I have some reservations against treating such bias as 'small'. It is substantial.
L102: 'demonstrate great' quality of TROPOMI XCH4 data' - see above. To be clear, it would be absolutely unfair to say that TROPOMI provides poor quality data, but it is an extremely challenging measurement and we should be careful and realistic about what can be achieved.
We agree that these values are not small. However, since they are within the mission requirements, we treated them as 'good' and described them as small. For the same reason, we describe them as being of great quality. To avoid confusion and be more cautious with our statement, we will slightly adapt the sentences as follows:
*..: they found a negative bias of -0.6% and -0.39% with TCCON $XCH_4$, respectively. These values are well within the mission requirements of 1.5% and therefore indicate a good quality of TROPOMI $XCH_4$ in this*

*part of China.*

L126: 'choice of flux inventory is likely important for...' – understatement, consider '...choice of flux inventory is critical for...'
This exact sentence was removed after restructuring the manuscript and moving the complete description of the sensitivity tests to the Appendix (see Sect. 1.1). In the main body of the manuscript we now say the following:
*These tests additionally include several anthropogenic emission inventories, because there is a wide range of global anthropogenic emission datasets available and these fluxes are crucial to simulate accurate concentrations in regions with large anthropogenic activity such as BTH.*
In a similar sentence in the Appendix, we now say:
*Sensitivity tests were carried out to identify a model configuration that matches the observations well. We have tested several physical parameterization schemes and anthropogenic fluxes because these elements are essential to accurately simulate tracer concentrations.*

L137: I recommend removing 3.2 as separate section (together with Table 1). Extract only information necessary for the manuscript (final config - Table 2) and put it in the description of "WRF-GHG modelling system". Move everything else to the appendix, where it fits better at the moment.
We thank the reviewer for the suggestion, it indeed fits better in the appendix. We have applied the changes as described above in Sect. 1.1.

L139: Consider relabeling test case E as BASE (or A).
Ok, adjusted.

L147: Where the anthropogenic fluxes also distributed vertically? Relevant since the discussed station is located near lots of anthropogenic activity. See: Brunner et al., 2019
We thank the reviewer for pointing out this interesting study. In our model simulations, all fluxes were released near the surface (in the lowest model layer). We will clarify this element by adding the following sentence in the section where we describe the model input data:
*All fluxes are released in the lowest model layer near the surface.*
As concluded by Brunner et al., 2019, the difference in near surface $CO_2$ is highest close to large point sources, such as power plants, and more significant in winter than in summer. More specifically, releasing the emissions too close to the ground possibly leads to a general overestimation of the near surface $CO_2$ in winter, while in summer this overestimation occurs mainly at night when the emissions are released within the stable nocturnal boundary layer instead of (possibly) above it. Since we have observed a small overestimation of in situ $CO_2$ at Xianghe at night, and there are some large point source in the region around the site, this emission simplification could explain some of the model-data discrepancies. We add this possible explanation to the new subsection on the discussion of the diurnal cycle of $CO_2$:
*Finally, all emissions in WRF-GHG are released near the surface in the lowest model layer which is a simplification of reality since especially industrial and energy sources (power plants) usually emit $CO_2$ at higher altitudes. Brunner et al., 2019 showed that neglecting this fact could lead to an overestimation of the near surface $CO_2$ concentration, which is larger in winter.*

L155: Please add info on TCCON measurement frequency for clarity.
We have added the following sentence in the data description section:
*Depending on the weather and measurement status, observations occur every 5-20 min.*

L188: Please provide info on the regridding method.
In our updated analysis (see Sect. 1.3), the weights are not used directly anymore. The TROPOMI a priori profiles and averaging kernels are regridded with the first order conservative remapping function of CDO to the WRF-GHG grid to enable calculating the smoothed $XCH_4$ values on the WRF-GHG grid. In this calculation, the smoothed $XCH_4$ will only be calculated for cells where there is information available in the (regridded) TROPOMI product. It is sufficient to take an average of all daily values over a specific period and ignoring the NaNs, to create a comparable map of the TROPOMI seasonal/yearly average (ignoring

model data at locations where there is no observation on a specific day).

L197: When comparing to TCCON, WRF output for methane was extended. Why not extend also here? At the very least this extension of 40 ppb could be added to the background (or as extra 'offset') to WRF numbers to avoid explaining the shifting of the colour scales when discussing each figure.
In the updated TROPOMI analysis, WRF-GHG profiles are extended with TROPOMI a priori profiles above 50 hPa (Sect. 1.3). As a result, all figures have the same color scale and a difference plot has been added.

L213-215: The 2 ppm bias is brushed over, and the reference to comparison CAMS-TCCON is missing. See also major comment no. 2.
L221: Again comparison CAMS-TCCON mentioned without reference. Numbers should be given, with uncertainty if possible.
As already stated in our reply on the shared concerns of both reviewers (Sect. 1.1), we have added two separate sections discussing the $CO_2$ and $CH_4$ biases, including the CAMS-TCCON comparison with the corresponding values. The new paragraph on the CAMS-TCCON comparison for $XCO_2$ is detailed in Sect. 1.2 above, while for $XCH_4$ it was further explained in our reply to your third major comment above.

L230: In previous section, the authors followed the order $CO_2$ - $CH_4$ - CO when discussing compounds. Now this order is reversed. Would make it easier to navigate if the ordering was consistent.
This has been corrected in the new structure of the revised manuscript.

L231: 'According to WRF-GHG'   'Based on our results'
Ok, adjusted.

L233: 'Energy sources and biomass burning are not important for the observations at Xianghe. Both residential and transportation tracers show larger values in winter, which is in agreement with higher emissions in that period of the year due to colder air temperatures.' - all these statements would use support from numbers (comment relevant throughout the manuscript).
We have added the median values of the main sector contributions in the corresponding paragraphs. Moreover, we added a large table displaying the median and interquartile range for all sectors.
More specifically for the subsection on CO, we suggest the following improvement:
  *Based on our model simulations, the CO column time series is primarily influenced by sources from the industry, residential and transportation sectors (see Fig. 4e). With a median value of 25.23 ppb, the industry sector is the largest contributor to XCO, followed by the residential and transportation sectors with values of 10.64 ppb and 10.54 ppb, respectively (see Table 4). Both residential and transportation tracers show larger values in winter, peaking in February with monthly median values of 33.03 ppb and 19.87 ppb. This increase is in agreement with higher emissions in that period of the year due to colder air temperatures (Guevara et al., 2021). Finally, energy sources and biomass burning are not important for the observations at Xianghe.*

L240: 'The main sectors contributing to the $CO_2$ data at Xianghe' - consider: 'The main sectors contributing to the modelled $CO_2$ variability at Xianghe'
Ok, adjusted.

L241: Note that $CO_2$ for VPRM model in WRF relies on MODIS surface classification and will predict zero fluxes over urban areas, while in fact $CO_2$ fluxes from urban biosphere can still be substantial. Unless VPRM was modified to handle these fluxes, this might have higher than anticipated impact if urban classification dominates in the vicinity of your station (in the model).
We have not modified the VPRM module to handle urban biospheric $CO_2$ fluxes. However, WRF-GHG is rather overestimating the $CO_2$ diurnal amplitude instead of underestimating, which would be the case when overlooking biospheric fluxes from within urban areas. Moreover, as added to the discussion on the diurnal $CO_2$ cycle (see Sect. 1.2), the urban classification is likely underestimated in the region.

L249: '...the year, which is in agreement with the general emissions patterns in China' - add a reference.
We have added a reference to Guevara et al., 2021 describing the monthly variation in the emission data set

used in this study.

L254-L256: Again, numbers needed to support the statements about emission contributions.
As stated above, this has been added in the revised manuscript. More specifically for $CH_4$:
*For $CH_4$, the simulated signal at Xianghe is mainly determined by three sectors: energy, residential & waste (which combines both residential heating and waste management sectors) and agriculture (Fig. 4c,d). They respectively contribute with a median enhancement of 11.13 ppb, 5.86 ppb and 4.75 ppb above the background for the columns and 48.86 ppb, 66.31 ppb and 49.79 ppb near the surface (see Table 1). Furthermore there is a small contribution from wetlands in summer, peaking in July with a median tracer contribution of 1.49 ppb for the columns and 10.65 ppb near the surface. Other sectors such as industry, transportation, termites and biomass burning seem to be irrelevant at Xianghe. ... There is a larger residential signal in winter, where the median tracer contribution peaks with 13.43 ppb in February for the columns and with 132.75 ppb in January, near the surface. Meanwhile, the influence from agriculture reaches its maximum in September (monthly median values of 14.46 ppb for $XCH_4$ and 196.07 ppb for in situ $CH_4$) and its minimum in March-April (monthly median values of 0.89 ppb for $XCH_4$ and 11.49 ppb for in situ $CH_4$). This corresponds with the seasonal pattern of emissions within CAMS-GLOB-ANT.*

L276-280: 'Unfortunately, the source...'. Maybe I misunderstand. The authors suggest that factors for $CH_4$ emissions from agriculture are constant - and yet there is a peak in emissions? That sounds like a bug in the emission preprocessing code. Please clarify.
As explained in the text, the issue is already present in the emission dataset (before preprocessing). The CAMS-GLOB-ANT v5.3 dataset can be downloaded upon registration from the ECCAD platform. This data shows a clear peak in livestock emissions in September over China. However, the corresponding documentation does not provide additional information on the origin of this variation. Given the unsuccessful communication with the creators regarding this matter, the basis of it remains unclear. In an attempt to clarify this in the manuscript, we propose to reorder some sentences to:
*According to CAMS-GLOB-ANT, the most important agriculture subsector in the region of the Xianghe site is livestock. The data set shows maximum livestock emissions in September in the wide region around Xianghe and minimum values in March and April. Unfortunately, the source of these monthly variations in $CH_4$ emissions within the CAMS-GLOB-ANT data set remains somewhat unclear, as the accompanying data set of temporal factors, CAMS-GLOB-TEMPO (Guevara et al., 2021), references constant factors for $CH_4$ emissions from agricultural sources.*

L286-289: 'In CAMS-GLOB-ANT, the waste sector is the most important one in the Xianghe region...' - numbers needed; also, why does this need support from Fig4, where direct emissions are available?
We have added numbers to this statement by calculating the sum of the monthly $CH_4$ emissions from CAMS-GLOB-ANT over all cells in the region between 38-41 °N and 115-119 °E:
*In CAMS-GLOB-ANT, the waste sector is the most important one in the Xianghe region and assumed to be relatively constant throughout the year: monthly total $CH_4$ emissions between 38-41 °N and 115-119 °E range between 0.0408 Tg and 0.0452 Tg. In summer, total residential combustion emissions in the region can be as low as 0.0039 Tg per month, while in winter, they are almost of the same size as the waste emissions: 0.0357 Tg.*
The reference to Fig 4c,d was given to restate that the model's tracer contributions of this sector indeed follow the same seasonality as the emissions. However, we agree that this information might be redundant so we propose to remove that sentence.

L299-300: '...ensemble against GOSAT observations by Parker et al. (2020), a general underestimation of the seasonal amplitude in China was found. This would mean an underestimation of the wetland CH4 emissions in summer.' - Needs clearer phrasing. As it stands the sentence is imprecise. Note that the underestimation of amplitude doesn't automatically translate into underestimation of wetland emissions.
While the underestimation of amplitude does not necessarily indicate an underestimation of wetland emissions, it remains a possibility. We removed this particular sentence, whereas this paragraph is now the following:
*In an evaluation of the WetCHARTs ensemble against GOSAT observations by Parker et al., 2020, a gen-*

Table 1: Statistics of the different tracer contributions above the background over the complete simulation period. Q1 and Q3 represent the first and third quartile, respectively, where 50 % of the data falls in between.

| | XCO$_2$ (ppm) | | | in situ CO$_2$ (ppm) | | | XCH$_4$ (ppb) | | | in situ CH$_4$ (ppb) | | | XCO (ppb) | | |
|---|---|---|---|---|---|---|---|---|---|---|---|---|---|---|---|
| | Q1 | median | Q3 | Q1 | median | Q3 | Q1 | median | Q3 | Q1 | median | Q3 | Q1 | median | Q3 |
| Biomass burning | 0.00 | 0.00 | 0.00 | 0.00 | 0.00 | 0.00 | 0.00 | 0.00 | 0.00 | 0.00 | 0.00 | 0.00 | 0.00 | 0.00 | 0.00 |
| Energy | 0.36 | 0.85 | 1.53 | 2.69 | 6.81 | 14.06 | 2.61 | 11.13 | 28.9 | 11.93 | 48.86 | 137.66 | 0.09 | 0.21 | 0.37 |
| Residential (& waste) | 0.03 | 0.06 | 0.17 | 0.31 | 0.69 | 2.15 | 2.65 | 5.86 | 10.72 | 31.24 | 66.31 | 124.8 | 4.75 | 10.64 | 20.45 |
| Industry | 0.24 | 0.63 | 1.14 | 2.64 | 5.74 | 10.67 | 0.07 | 0.17 | 0.30 | 0.76 | 1.65 | 3.02 | 11.83 | 25.23 | 45.39 |
| Transportation | 0.08 | 0.16 | 0.25 | 0.80 | 1.72 | 3.18 | 0.06 | 0.12 | 0.20 | 0.65 | 1.39 | 2.57 | 5.57 | 10.54 | 17.48 |
| Biosphere | -0.77 | 0.04 | 0.31 | 0.05 | 2.59 | 7.50 | | | | | | | | | |
| Ocean | -0.00 | -0.00 | -0.00 | -0.01 | -0.00 | -0.00 | | | | | | | | | |
| Agriculture | | | | | | | 2.00 | 4.75 | 9.49 | 23.65 | 49.79 | 96.06 | | | |
| Wetlands | | | | | | | 0.02 | 0.12 | 0.62 | 0.07 | 0.51 | 2.95 | | | |
| Termites | | | | | | | 0.17 | 0.29 | 0.46 | 1.13 | 2.03 | 3.22 | | | |

[Figure]

Figure G: Monthly mean tracer contributions above the background for (a) XCO$_2$, (b) XCH$_4$, (c) XCO, (d) in situ CO$_2$ and (e) in situ CH$_4$ simulated concentrations at Xianghe.

*eral underestimation of the seasonal amplitude in China was found. Furthermore, Chen et al., 2022 showed increased posterior wetlands emissions compared to the a priori values when inferring yearly $CH_4$ emissions over China using TROPOMI satellite observations. This could point to an underestimation of the wetland emissions in the current study.*

L325: 'More specifically, we looked at the daily mean column concentrations above the background for every wind direction...' Daily means from time when observations were available, correct?
Indeed, only model values for times where there are corresponding observations are used for this calculation.

L347-L365: While the importance of PBL height is paramount to the correct interpretation of in situ data, this section doesn't bring a lot of insight into the discussion. It could be quite illuminating if expanded.
We have added more quantitative values to this section (see two comments below) and additionally expanded the discussion on the model nighttime overestimation of the $CO_2$ concentration, see Sect. 1.2 above.

L352: 'This stable nocturnal layer is quite shallow and...' - how shallow? Numbers needed.
According to the WRF-GHG model simulations the stable nocturnal layer (defined between 18:00 and 07:00 LT) has a median height of 117.25 m, with an interquartile range of 49.13 m - 429.89 m. We have added this in the manuscript:
*Right after sunset, the height of the PBL drops to its lowest value (≈ 50 m - 430 m in WRF-GHG), ...*

L361: 'Remark that WRF-GHG is very well capable at simulating this diurnal variation of both $CO_2$ and $CH_4$ in situ observations.' - Again, quantitative analysis would be more than welcome. Also: 'Remark that' 'Note that'
We have adjusted this paragraph by including a more quantitative analysis:
*Figure 6 shows the diurnal variation of the PBL height as simulated by WRF-GHG and the $CO_2$ and $CH_4$ concentrations near the surface (both simulated and observed). Indeed, the height of the PBL in WRF-GHG is largest in the afternoon when solar radiation is strongest, reaching its peak at 15:00 (local time). This corresponds with the lowest simulated surface concentrations (Fig. 6b,c), where we find median (and interquartile) values of 419.41 (413.75 - 428.69) ppm for $CO_2$ and 2043.15 (1985.78 - 2164.26) ppb for $CH_4$. Right after sunset, the height of the PBL drops to its lowest value (≈ 50 m - 430 m in WRF-GHG), after which it persists during the course of the night, until sunrise. This period corresponds with slightly increasing $CO_2$ and $CH_4$ concentrations as emissions near the surface accumulate within this stable shallow layer. Hence, the highest concentrations are found in the early morning: at 3:00 for $CO_2$ with a value of 445.91 (430.03 - 461.75) ppm and at 8:00 for $CH_4$ with 2240.29 (2066.31 - 2484.09) ppb. Remark however, that the simulated $CO_2$ concentrations remain quite stable between 3:00 and 8:00. As the PBL height starts to rise at 8:00 due to turbulent mixing, both $CO_2$ and $CH_4$ start to drop in WRF-GHG, creating a diurnal cycle.*
*Note that WRF-GHG is quite capable at simulating this diurnal variation of both $CO_2$ and $CH_4$ in situ observations. For $CO_2$, we find a small underestimation during the day as the lowest concentrations in the observations generally occur at 16:00 with a median (and interquartile) value of 421.13 (415.62 - 431.56) ppm, and a larger overestimation at night, where the observations peak at 7:00 with 443.21 (427.81 - 459.10) ppm. This leads to an overestimation of the $CO_2$ diurnal amplitude in WRF-GHG with 4.42 ppm. For $CH_4$, the observations show minimal concentrations at 16:00 with a median (and interquartile) value of 2041.84 (1982.65 – 2135.88) ppb, which are well captured by WRF-GHG, even though one hour earlier. The peak $CH_4$ concentrations however, are observed at 6:00 with a median (and interquartile) value of 2252.71 (2104.36 – 2428.02) ppb, portraying a small model underestimation. Together, this leads to a small underestimation of the $CH_4$ diurnal amplitude in WRF-GHG of 13.63 ppb.*

L372: 'enhancement (sum of all WRF-GHG tracers above the background) is smaller' - delete fragment in parentheses (redundant)
Ok, part between parentheses is removed.

L381: 'The highest values overall...' - how high?
As can be seen from the colorbar in Fig. 9g, hourly values up 400 ppb or more are found for the Energy tracer at night. We add this information between brackets:

*The highest values overall (> 400 ppb) are found for the energy tracer at night and they are coming from the east, where ...*

L382: '...are coming from the east. To the east are...'   '...are coming from the east, where...'
This has been adapted.

L384-385: Concluding the cause before proof is presented. Overestimation of coal-mine emissions cannot be argued before next section, where TROPOMI data is used as extra argument. By this point of the text also bias in the night-time PBL height can explain the observed $CH_4$ in situ bias.
In these referred lines, we state that we have observed a model overestimation together with large energy contributions that mainly occur from the east at night. We then present our hypothesis that it might be caused by an overestimation of the coal-mine emissions. The next section (new Sect. 4.4) then aims to assess this statement with TROPOMI data. In our opinion, the proof is not presented before the cause: we observe a distinctive pattern that we subsequently try to explain. To clarify this, we slightly adapted lines 386-387 as follows:
*In the Discussion section we further investigate this hypothesis by comparing WRF-GHG concentration fields with TROPOMI observations.*
Furthermore, it is unlikely that the bias is purely caused by incorrect nighttime PBL height since the observed overestimation is mainly for the Energy tracer.

L388: Section 'Assessing CH4 emission sources' is very interesting but sadly mostly based on subjective visual analysis of the figures. More quantitative results should be presented to support the conclusions.
The revised manuscript now contains a quantitative analysis of the subject, as elaborated in Sect. 1.3.

L396-L403: Please briefly expand here. I assume this is a mixture of cloud + albedo effects. Is there enough confidence that the issues causing missing TROPOMI data are completely gone in the areas immediately neighbouring the "white spots" on the map in Fig12?
To clarify what I mean: it is clear that the highest discrepancy is present immediately to the east of 114 deg. E, where many of the gridded cells with those high values immediately neighbour the area of missing TROPOMI data. If the filtration procedure used for TROPOMI is not 100 % sure, then the discrepancy might be an artefact rather than real signal. I would appreciate if this can be addressed.
Most of the pixels that are filtered out in the TROPOMI product are because of orography or clouds. It has indeed been described in the S5P validation report (which is published https://mpc-vdaf.tropomi.eu/) that the $XCH_4$ product often shows artificially high values near strong surface gradients, such as coastal or mountain regions. The lines which the reviewer is pointing at however, are related to WRF-GHG model simulations where the link between elevated $XCH_4$ values and emissions is evident as can be seen in Fig 12a and b. Nevertheless, the high values east of 114° on the TROPOMI map might indeed be related to the inaccurate filter. Since the WRF-GHG simulations also show increased $XCH_4$ values in that region, it is more likely however that this enhancement is a mixture of both a true signal and an inaccurate retrieval due to the mountain range.

L406: Why not add 40 ppb as offset and use same scales? Would be easier to interpret. See comment above.
As replied to the comment above, we have included an updated comparison with TROPOMI by extending the WRF-GHG profiles above 50 hPa.

L421: In some sections results from 9 x 9 km2 are used for comparisons.
In all analysis, the 3 km results where used, expect for the TROPOMI comparison. This is to have a wider region covered by the model. Moreover, simulations from d02 are also closer to the TROPOMI resolution. We will clarify this by adding *ground-based observations* in that particular sentence.
Further, we have added the following sentence in the subsection describing how WRF-GHG is compared with TROPOMI:
*Note that we use the model simulations from d02 (which has a horizontal resolution of $9\times9$ $km^2$) for this analysis, instead of d03 as for the comparisons with Xianghe observations, since a larger spatial extent is advantageous for an effective comparison with TROPOMI.*

L430: '...a small underestimation of -1.43 ppm and -3.03 ppb with respect to TCCON $XCO_2$ and $XCH_4$, respectively.' Please clarify that these are averages. Add uncertainties.
We have clarified in the sentence that these values are mean bias errors:
*A negligible bias was found for XCO while WRF-GHG showed a small underestimation with a mean bias error of -1.43 ppm and -3.03 ppb with respect to TCCON $XCO_2$ and $XCH_4$, respectively. The in situ time series of $CO_2$ and $CH_4$ are slightly overestimated by WRF-GHG, with a mean bias error of 2.52 ppm and 9.43 ppb, respectively.*

L431: 'slightly overestimated' - I don't think 'slightly' fits here, these numbers are substantial (again, to be clear: I do not mean the results are bad)
As replied to the comment above, we have clarified that these numbers refer to mean bias error values. Additionally we have replaced 'slightly overestimated' with 'somewhat overestimated'.

L438: 'This difference is likely because of the lack of strong photo-synthetically active vegetation in the neighborhood of Xianghe.' - in the model or physical world? Or both?
Both. Even though we argued that the land cover classification inside the model might be inaccurate (more cropland than built-up surfaces near the site), this statement refers to the lack of mainly forests nearby, as this vegetation class shows the strongest $CO_2$ uptake due to photosynthesis. The main forested area in the region is found in the mountain range north and west of Beijing, at a distance of at least 50 km from Xianghe. This can be seen on public mapping portals like Google Earth or the Copernicus Dynamic Land Cover Map (link) and is correctly represented in WRF-GHG.

L439: 'For CH4...'   'On average, for CH4,...'
Ok, adjusted.

L453-454: 'peak values are found in the early morning around sunrise when local emissions are strongest...' - Please clarify. Not all the compounds have strongest emissions in the morning ($CH_4$ for example).
Indeed, it is correct that not all emissions peak in the morning. We will correct this sentence by omitting the reference to the emissions as the PBL variation has the strongest influence on the diurnal variation of the in situ concentrations.
*On the other hand, peak values are found in the early morning around sunrise when species have been accumulated all night in the lowest layer of the atmosphere.*

Table 3.: Consider reordering columns, having 'This study' as first for easier reader experience.
Ok, done.

Figure 7: It appears that the fonts in this figure are smaller, but maybe it's my eyesight. Please check. Also, consider using $\Delta XCO2$ and $\Delta CO2$ as axis labels.
As reaction to a comment from the other reviewer, this figure was omitted from the manuscript.

**References**

Agustí-Panareda, A., Barré, J., Massart, S., Inness, A., Aben, I., Ades, M., Baier, B. C., Balsamo, G., Borsdorff, T., Bousserez, N., Boussetta, S., Buchwitz, M., Cantarello, L., Crevoisier, C., Engelen, R., Eskes, H., Flemming, J., Garrigues, S., Hasekamp, O., ... Wu, L. (2023). Technical note: The CAMS greenhouse gas reanalysis from 2003 to 2020 [Publisher: Copernicus GmbH]. *Atmospheric Chemistry and Physics*, *23*(6), 3829–3859. https://doi.org/10.5194/acp-23-3829-2023

Apituley, A., Pedergnana, M., Sneep, M., Veefkind, P. J., Loyola, D., Hasekamp, O., Lorente Delgado, A., & Borsdorff, T. (2023, September 26). Sentinel-5 precursor/TROPOMI level 2 product user manual methane [ref: SRON-S5P-LEV2-MA-001, issue: 2.6.0].

Brunner, D., Kuhlmann, G., Marshall, J., Clément, V., Fuhrer, O., Broquet, G., Löscher, A., & Meijer, Y. (2019). Accounting for the vertical distribution of emissions in atmospheric $CO_2$ simulations [Publisher: Copernicus GmbH]. *Atmospheric Chemistry and Physics*, *19*(7), 4541–4559. https://doi.org/10.5194/acp-19-4541-2019

Butler, M. P., Lauvaux, T., Feng, S., Liu, J., Bowman, K. W., & Davis, K. J. (2020). Atmospheric simulations of total column CO2 mole fractions from global to mesoscale within the carbon monitoring system flux inversion framework [Number: 8 Publisher: Multidisciplinary Digital Publishing Institute]. *Atmosphere*, *11*(8), 787. https://doi.org/10.3390/atmos11080787

Callewaert, S., Brioude, J., Langerock, B., Duflot, V., Fonteyn, D., Müller, J.-F., Metzger, J.-M., Hermans, C., Kumps, N., Ramonet, M., Lopez, M., Mahieu, E., & De Mazière, M. (2022). Analysis of $CO_2$, $CH_4$, and CO surface and column concentrations observed at réunion island by assessing WRF-chem simulations [Publisher: Copernicus GmbH]. *Atmospheric Chemistry and Physics*, *22*(11), 7763–7792. https://doi.org/10.5194/acp-22-7763-2022

Che, K., Cai, Z., Liu, Y., Wu, L., Yang, D., Chen, Y., Meng, X., Zhou, M., Wang, J., Yao, L., & Wang, P. (2022). Lagrangian inversion of anthropogenic CO2 emissions from beijing using differential column measurements [Publisher: IOP Publishing]. *Environmental Research Letters*, *17*(7), 075001. https://doi.org/10.1088/1748-9326/ac7477

Chen, Z., Jacob, D. J., Nesser, H., Sulprizio, M. P., Lorente, A., Varon, D. J., Lu, X., Shen, L., Qu, Z., Penn, E., & Yu, X. (2022). Methane emissions from china: A high-resolution inversion of TROPOMI satellite observations [Publisher: Copernicus GmbH]. *Atmospheric Chemistry and Physics*, *22*(16), 10809–10826. https://doi.org/10.5194/acp-22-10809-2022

Dekker, I. N., Houweling, S., Aben, I., Röckmann, T., Krol, M., Martínez-Alonso, S., Deeter, M. N., & Worden, H. M. (2017). Quantification of CO emissions from the city of madrid using MOPITT satellite retrievals and WRF simulations. *Atmospheric Chemistry and Physics*, *17*(23), 14675–14694. https://doi.org/10.5194/acp-17-14675-2017

Díaz-Isaac, L. I., Lauvaux, T., & Davis, K. J. (2018). Impact of physical parameterizations and initial conditions on simulated atmospheric transport and $CO_2$ mole fractions in the US midwest [Publisher: Copernicus GmbH]. *Atmospheric Chemistry and Physics*, *18*(20), 14813–14835. https://doi.org/10.5194/acp-18-14813-2018

Dong, X., Yue, M., Jiang, Y., Hu, X.-M., Ma, Q., Pu, J., & Zhou, G. (2021). Analysis of $CO_2$ spatio-temporal variations in china using a weather–biosphere online coupled model [Publisher: Copernicus GmbH]. *Atmospheric Chemistry and Physics*, *21*(9), 7217–7233. https://doi.org/10.5194/acp-21-7217-2021

Feng, S., Lauvaux, T., Newman, S., Rao, P., Ahmadov, R., Deng, A., Díaz-Isaac, L. I., Duren, R. M., Fischer, M. L., Gerbig, C., Gurney, K. R., Huang, J., Jeong, S., Li, Z., Miller, C. E., O'Keeffe, D., Patarasuk, R., Sander, S. P., Song, Y., . . . Yung, Y. L. (2016). Los angeles megacity: A high-resolution land–atmosphere modelling system for urban CO emissions. *Atmospheric Chemistry and Physics*, *16*(14), 9019–9045. https://doi.org/10.5194/acp-16-9019-2016

Gerbig, C., Lin, J. C., Wofsy, S. C., Daube, B. C., Andrews, A. E., Stephens, B. B., Bakwin, P. S., & Grainger, C. A. (2003). Toward constraining regional-scale fluxes of CO2 with atmospheric observations over a continent: 1. observed spatial variability from airborne platforms. *Journal of Geophysical Research: Atmospheres*, *108*. https://doi.org/10.1029/2002JD003018

Guevara, M., Jorba, O., Tena, C., Denier van der Gon, H., Kuenen, J., Elguindi, N., Darras, S., Granier, C., & Pérez García-Pando, C. (2021). Copernicus atmosphere monitoring service TEMPOral profiles (CAMS-TEMPO): Global and european emission temporal profile maps for atmospheric chemistry modelling [Publisher: Copernicus GmbH]. *Earth System Science Data*, *13*(2), 367–404. https://doi.org/10.5194/essd-13-367-2021

Gupta, S., Tiwari, Y. K., Revadekar, J. V., Burman, P. K. D., Chakraborty, S., & Gnanamoorthy, P. (2021). An intensification of atmospheric CO2 concentrations due to the surface temperature extremes in india. *Meteorology and Atmospheric Physics*, *133*(6), 1647–1659. https://doi.org/10.1007/s00703-021-00834-w

Hu, C., Liu, S., Wang, Y., Zhang, M., Xiao, W., Wang, W., & Xu, J. (2018). Anthropogenic CO2 emissions from a megacity in the yangtze river delta of china. *Environmental Science and Pollution Research*, *25*(23), 23157–23169. https://doi.org/10.1007/s11356-018-2325-3

Hu, X.-M., Crowell, S., Wang, Q., Zhang, Y., Davis, K. J., Xue, M., Xiao, X., Moore, B., Wu, X., Choi, Y., & DiGangi, J. P. (2020). Dynamical downscaling of CO2 in 2016 over the contiguous united states using WRF-VPRM, a weather-biosphere-online-coupled model. *Journal of Advances in Modeling Earth Systems*, *12*(4), e2019MS001875. https://doi.org/10.1029/2019MS001875

Hu, X.-M., Gourdji, S. M., Davis, K. J., Wang, Q., Zhang, Y., Xue, M., Feng, S., Moore, B., & Crowell, S. M. R. (2021). Implementation of improved parameterization of terrestrial flux in WRF-VPRM improves the simulation of nighttime CO2 peaks and a daytime CO2 band ahead of a cold front. *Journal of Geophysical Research: Atmospheres*, *126*(10), e2020JD034362. https://doi.org/10.1029/2020JD034362

Jung, M., Henkel, K., Herold, M., & Churkina, G. (2006). Exploiting synergies of global land cover products for carbon cycle modeling. *Remote Sensing of Environment*, *101*(4), 534–553. https://doi.org/https://doi.org/10.1016/j.rse.2006.01.020

Lac, C., Donnelly, R. P., Masson, V., Pal, S., Riette, S., Donier, S., Queguiner, S., Tanguy, G., Ammoura, L., & Xueref-Remy, I. (2013). CO$_2$ dispersion modelling over paris region within the CO$_2$-MEGAPARIS project [Publisher: Copernicus GmbH]. *Atmospheric Chemistry and Physics*, *13*(9), 4941–4961. https://doi.org/10.5194/acp-13-4941-2013

Laughner, J. L., Roche, S., Kiel, M., Toon, G. C., Wunch, D., Baier, B. C., Biraud, S., Chen, H., Kivi, R., Laemmel, T., McKain, K., Quéhé, P.-Y., Rousogenous, C., Stephens, B. B., Walker, K., & Wennberg, P. O. (2023). A new algorithm to generate a priori trace gas profiles for the GGG2020 retrieval algorithm [Publisher: Copernicus GmbH]. *Atmospheric Measurement Techniques*, *16*(5), 1121–1146. https://doi.org/10.5194/amt-16-1121-2023

Lauvaux, T., Miles, N. L., Deng, A., Richardson, S. J., Cambaliza, M. O., Davis, K. J., Gaudet, B., Gurney, K. R., Huang, J., O'Keefe, D., Song, Y., Karion, A., Oda, T., Patarasuk, R., Razlivanov, I., Sarmiento, D., Shepson, P., Sweeney, C., Turnbull, J., & Wu, K. (2016). High-resolution atmospheric inversion of urban CO2 emissions during the dormant season of the indianapolis flux experiment (INFLUX). *Journal of Geophysical Research: Atmospheres*, *121*(10), 5213–5236. https://doi.org/10.1002/2015JD024473

Li, X., Hu, X.-M., Cai, C., Jia, Q., Zhang, Y., Liu, J., Xue, M., Xu, J., Wen, R., & Crowell, S. M. R. (2020). Terrestrial CO2 fluxes, concentrations, sources and budget in northeast china: Observational and modeling studies. *Journal of Geophysical Research: Atmospheres*, *125*(6), e2019JD031686. https://doi.org/10.1029/2019JD031686

Liu, Y., Yue, T., Zhang, L., Zhao, N., Zhao, M., & Liu, Y. (2018). Simulation and analysis of XCO2 in north china based on high accuracy surface modeling. *Environmental Science and Pollution Research International*, *25*(27), 27378–27392. https://doi.org/10.1007/s11356-018-2683-x

Mahadevan, P., Wofsy, S. C., Matross, D. M., Xiao, X., Dunn, A. L., Lin, J. C., Gerbig, C., Munger, J. W., Chow, V. Y., & Gottlieb, E. W. (2008). A satellite-based biosphere parameterization for net ecosystem CO2 exchange: Vegetation photosynthesis and respiration model (VPRM). *Global Biogeochemical Cycles*, *22*(2). https://doi.org/10.1029/2006GB002735

Park, C., Gerbig, C., Newman, S., Ahmadov, R., Feng, S., Gurney, K. R., Carmichael, G. R., Park, S.-Y., Lee, H.-W., Goulden, M., Stutz, J., Peischl, J., & Ryerson, T. (2018). CO2 transport, variability, and budget over the southern california air basin using the high-resolution WRF-VPRM model during the CalNex 2010 campaign. *Journal of Applied Meteorology and Climatology*, *57*(6), 1337–1352. https://doi.org/10.1175/JAMC-D-17-0358.1

Park, C., Park, S.-Y., Gurney, K. R., Gerbig, C., DiGangi, J. P., Choi, Y., & Lee, H. W. (2020). Numerical simulation of atmospheric CO2 concentration and flux over the korean peninsula using WRF-VPRM model during korus-AQ 2016 campaign [Publisher: Public Library of Science]. *PLOS ONE*, *15*(1), e0228106. https://doi.org/10.1371/journal.pone.0228106

Parker, R. J., Wilson, C., Bloom, A. A., Comyn-Platt, E., Hayman, G., McNorton, J., Boesch, H., & Chipperfield, M. P. (2020). Exploring constraints on a wetland methane emission ensemble (WetCHARTs) using GOSAT observations [Publisher: Copernicus GmbH]. *Biogeosciences*, *17*(22), 5669–5691. https://doi.org/10.5194/bg-17-5669-2020

Pillai, D., Gerbig, C., Ahmadov, R., Rödenbeck, C., Kretschmer, R., Koch, T., Thompson, R., Neininger, B., & Lavrié, J. V. (2011). High-resolution simulations of atmospheric CO2 over complex terrain –

representing the ochsenkopf mountain tall tower. *Atmospheric Chemistry and Physics*, *11*(15), 7445–7464. https://doi.org/10.5194/acp-11-7445-2011

Ramonet, M., Ciais, P., Apadula, F., Bartyzel, J., Bastos, A., Bergamaschi, P., Blanc, P. E., Brunner, D., Caracciolo di Torchiarolo, L., Calzolari, F., Chen, H., Chmura, L., Colomb, A., Conil, S., Cristofanelli, P., Cuevas, E., Curcoll, R., Delmotte, M., di Sarra, A., . . . Yver Kwok, C. (2020). The fingerprint of the summer 2018 drought in europe on ground-based atmospheric CO2 measurements [Publisher: Royal Society]. *Philosophical Transactions of the Royal Society B: Biological Sciences*, *375*(1810), 20190513. https://doi.org/10.1098/rstb.2019.0513

Ramonet, M., Langerock, B., Warneke, T., & Eskes, H. (2021). Validation report of the CAMS greenhouse gas global reanalysis, years 2003-2020 [Publisher: [object Object]]. https://doi.org/10.24380/438C-4597

Schulzweida, U. (2020). *Climate data operators (CDO) user guide* (Version 2.3.0).

Sha, M. K., Langerock, B., Blavier, J.-F. L., Blumenstock, T., Borsdorff, T., Buschmann, M., Dehn, A., De Mazière, M., Deutscher, N. M., Feist, D. G., García, O. E., Griffith, D. W. T., Grutter, M., Hannigan, J. W., Hase, F., Heikkinen, P., Hermans, C., Iraci, L. T., Jeseck, P., . . . Zhou, M. (2021). Validation of methane and carbon monoxide from sentinel-5 precursor using TCCON and NDACC-IRWG stations [Publisher: Copernicus GmbH]. *Atmospheric Measurement Techniques*, *14*(9), 6249–6304. https://doi.org/10.5194/amt-14-6249-2021

Tian, Y., Hong, X., Shan, C., Sun, Y., Wang, W., Zhou, M., Wang, P., Lin, P., & Liu, C. (2022). Investigating the performance of carbon monoxide and methane observations from sentinel-5 precursor in china [Number: 23 Publisher: Multidisciplinary Digital Publishing Institute]. *Remote Sensing*, *14*(23), 6045. https://doi.org/10.3390/rs14236045

*TROPOMI Level 2 Methane Total Column products. Version 02.* (2021). European Space Agency. https://doi.org/10.5270/S5P-3lcdqiv
    Copernicus Sentinel-5P (processed by ESA).

Xu, H., Xiao, J., & Zhang, Z. (2020). Heatwave effects on gross primary production of northern mid-latitude ecosystems [Publisher: IOP Publishing]. *Environmental Research Letters*, *15*(7), 074027. https://doi.org/10.1088/1748-9326/ab8760

Yang, Y., Zhou, M., Langerock, B., Sha, M. K., Hermans, C., Wang, T., Ji, D., Vigouroux, C., Kumps, N., Wang, G., De Mazière, M., & Wang, P. (2020). New ground-based fourier-transform near-infrared solar absorption measurements of $XCO_2$, $XCH_4$ and XCO at xianghe, china [Publisher: Copernicus GmbH]. *Earth System Science Data*, *12*(3), 1679–1696. https://doi.org/10.5194/essd-12-1679-2020

Yang, Y., Zhou, M., Wang, T., Yao, B., Han, P., Ji, D., Zhou, W., Sun, Y., Wang, G., & Wang, P. (2021). Spatial and temporal variations of $CO_2$ mole fractions observed at beijing, xianghe, and xinglong in north china [Publisher: Copernicus GmbH]. *Atmospheric Chemistry and Physics*, *21*(15), 11741–11757. https://doi.org/10.5194/acp-21-11741-2021

Zhao, X., Marshall, J., Hachinger, S., Gerbig, C., Frey, M., Hase, F., & Chen, J. (2019). Analysis of total column $CO_2$ and $CH_4$ measurements in berlin with WRF-GHG [Publisher: Copernicus GmbH]. *Atmospheric Chemistry and Physics*, *19*(17), 11279–11302. https://doi.org/10.5194/acp-19-11279-2019